# Concomitant activation of GLI1 and Notch1 contributes to racial disparity of human triple negative breast cancer progression

**Sumit Siddharth[1], Sheetal Parida[1], Nethaji Muniraj[1], Shawn Hercules[2], David Lim[3], Arumugam Nagalingam[1], Chenguang Wang[3], Balazs Gyorffy[4,5], Juliet M Daniel[2], Dipali Sharma[1]***

[1]Dept. of Oncology, Johns Hopkins University School of Medicine and Sidney Kimmel Comprehensive Cancer Center at Johns Hopkins, Baltimore, United States; [2]Department of Biology, MacMaster University, Hamilton, Canada; [3]Division of Biostatistics and Bioinformatics, Sidney Kimmel Comprehensive Cancer Center at Johns Hopkins, Baltimore, United States; [4]MTA TTK Momentum Cancer Biomarker Research Group, Budapest, Hungary; [5]Semmelweis University, Department of Bioinformatics and 2nd Dept. of Pediatrics, Budapest, Hungary

**Abstract** Mortality from triple negative breast cancer (TNBC) is significantly higher in African American (AA) women compared to White American (WA) women emphasizing ethnicity as a major risk factor; however, the molecular determinants that drive aggressive progression of AA-TNBC remain elusive. Here, we demonstrate for the first time that AA-TNBC cells are inherently aggressive, exhibiting elevated growth, migration, and cancer stem-like phenotype compared to WA-TNBC cells. Meta-analysis of RNA-sequencing data of multiple AA- and WA-TNBC cell lines shows enrichment of GLI1 and Notch1 pathways in AA-TNBC cells. Enrichment of GLI1 and Notch1 pathway genes was observed in AA-TNBC. In line with this observation, analysis of TCGA dataset reveals a positive correlation between GLI1 and Notch1 in AA-TNBC and a negative correlation in WA-TNBC. Increased nuclear localization and interaction between GLI1 and Notch1 is observed in AA-TNBC cells. Of importance, inhibition of GLI1 and Notch1 synergistically improves the efficacy of chemotherapy in AA-TNBC cells. Combined treatment of AA-TNBC-derived tumors with GANT61, DAPT, and doxorubicin/carboplatin results in significant tumor regression, and tumor-dissociated cells show mitigated migration, invasion, mammosphere formation, and CD44$^+$/CD24$^-$ population. Indeed, secondary tumors derived from triple-therapy-treated AA-TNBC tumors show diminished stem-like phenotype. Finally, we show that TNBC tumors from AA women express significantly higher level of GLI1 and Notch1 expression in comparison to TNBC tumors from WA women. This work sheds light on the racial disparity in TNBC, implicates the GLI1 and Notch1 axis as its functional mediators, and proposes a triple-combination therapy that can prove beneficial for AA-TNBC.

## Editor's evaluation

This paper represents a fundamental contribution to our understanding of potential differences in the incidence of triple negative breast cancer in Caucasian and African American women. In addition, the findings provide potential insights relating to the relative effectiveness of therapy in these populations, and suggest novel therapeutic approaches.

*For correspondence:
dsharma7@jhmi.edu

Competing interest: The authors declare that no competing interests exist.

DOI: https://doi.org/10.7554/eLife.70729

## Introduction

While breast cancer remains the most common cancer among women worldwide, mortality associated with breast cancer has been declining owing to the development of novel therapeutics and ensuing advances in clinical care. A large decline in mortality for female breast cancer (by ~40% since 1989) has been observed, but a mortality gap exists between White American (WA) and African American (AA) women (*Siegel et al., 2020*). Despite lower incidence rates (126.7/100,000 for AA vs. 130.8/100,000 for WA), AA women are 42% more likely to succumb to breast cancer-related mortality (*DeSantis et al., 2019*; *Siegel et al., 2020*). The difference in breast cancer incidence and mortality for AA and WA women is most prevalent in younger age-groups and gradually converges in higher age-groups. AA women younger than 50 have 1.9–2.6 times higher breast cancer-related mortality compared to WA women, which declines to 1.1–1.2 times higher in 70–89 age-group (*DeSantis et al., 2019*; *Siegel et al., 2020*).

Higher breast cancer-related mortality in AA women can be partly explained by the disproportionately higher incidence rates for triple negative breast cancer (TNBC). AA women have twice as high (38/100,000) TNBC incidence in comparison to WA women (19/100,000). TNBC is a very aggressive subtype of breast cancer characterized by the absence of estrogen receptor (ER), progesterone receptor (PR), and HER2 receptor expression. Not only do AA women have higher TNBC incidence, but the survival rate for TNBC is also significantly lower in AA women in comparison to WA women (5-year relative survival of only 14% for AA in comparison to 36% for WA). Factors such as increased prevalence of obesity and comorbidities; late stage at diagnosis; aggressive tumor characteristics; and reduced access to timely prevention, early detection, and high-quality treatment may contribute to higher TNBC-related mortality in AA women. It is important to note that this disparity in TNBC-related survival in AA women vs. WA women is evident even after adjusting for socioeconomic status and access to medical care, strongly indicating a biological basis (*Carey et al., 2006*; *Dietze et al., 2015*).

Most of the studies examining gene expression distinctions in AA and WA breast tumors have identified intriguing patterns according to ethnicities/race but their association with clinical outcomes and different subtypes of breast cancer is largely unexplored. Comparative assessment of the genomic landscape of breast tumors from AA and WA women reveals that AA tumors show higher prevalence of *TP53* mutations (42.9% vs. 27.6%) and lower level of *PIK3CA* mutations (20.0% vs. 33.9%) in comparison WA tumors. Greater intratumoral genetic heterogeneity including an enrichment of immortalization signature gene set is also observed in AA tumors (*Andey et al., 2020*; *Keenan et al., 2015*). Even after matching tumors for pathological characteristics, distinct differences in multiple genes including CRYBB2, PSPHL, and SOS1 have been observed in AA vs. WA invasive breast cancer (*Field et al., 2012*). Several unique gene expression variations in breast tumors as well as tumor stroma are noted in AA that may support higher microvessel density and macrophage infiltration (*Martin et al., 2009*; *Stewart et al., 2013*). While several cell cycle regulators such as *CDKN2A (p16)*, *CCNA2*, *CCNB1*, and *CCNE2* express at a higher level in AA breast tumors, a unique gene set comprising of PSPHL, CXCL10, CXCL11, ISG20, and GMDS is overexpressed in AA tumor stroma (*Grunda et al., 2012*; *Martin et al., 2009*). It is intuitive that unique genetic profiles observed in AA vs. WA breast tumors may vary furthermore according to the subtype of breast cancer, hence appropriate subtype stratification is required. Despite higher prevalence of TNBC and poorer TNBC-related survival among AA women, very few studies have attempted to examine the molecular phenotype of AA-TNBC in direct comparison to WA-TNBC. No significant differences are reported in somatic mutations between AA-TNBC and WA-TNBC in a customized set of 151 genes (*Omilian et al., 2020*) but elevated expression of VEGF-activated genes and proliferation-associated gene signature, altered TP53, NFB1, and AKT pathways and increased microvessel density has been associated with AA-TNBC (*Davis et al., 2020*; *Lindner et al., 2013*). While TNBC biology that drives the aggressive progression of AA-TNBC is still imprecise, it is clear that irrespective of stage at diagnosis, AA-TNBC is more aggressive with higher metastasis and a poorer survival than WA-TNBC.

In the present study, we sought to investigate the intrinsic differences in the molecular networks that may contribute to the racial disparity in TNBC progression in AA and WA women. We observed that AA-TNBC cells possess inherently higher migration and invasion potential along with elevated self-renewal potential and stem-like phenotype compared to WA-TNBC cells. Our study revealed an enrichment of *glioma-associated oncogene homolog 1* (GLI1) and Notch1 pathways in AA-TNBC as the key node and we found a positive correlation between GLI1 and Notch1 in AA-TNBC tumors in

contrast to WA-TNBC tumors. Inhibiting GLI1 and Notch1 potentiates the efficacy of chemotherapy regimens in AA-TNBC and results in mitigation of stemness phenotype.

## Results

### AA-TNBC cells possess distinct functional alterations to support aggressive tumor progression

With an aim to investigate the racial disparity in TNBC, we started this investigation by analyzing the SEER (The Surveillance, Epidemiology, and End Results Program) data specifically focusing on differences between AA and WA women. Breast cancer incidence rates revealed higher incidences of TNBC among black women (including Hispanics) than white women (including Hispanics) over multiple years and several age-groups (*Figure 1A and B*). Percentage of TNBC incidence among black women was twice as high as white women (*Figure 1D*). The 5-year relative survival rate analysis for TNBC patients based on race/ethnicity and age showed a higher percentage of surviving population among white women (including Hispanics) than black women (including Hispanics) in all the age-groups (15–39, 40–64, 65–74, 75+ ages) (*Figure 1C*). Black women showed higher prevalence of grade 3 TNBC at the time of diagnosis compared to white women (83.1% vs. 77.98%) in SEER dataset (*Figure 1E*). Analysis of the TCGA data indicated higher percentage of AA-TNBC (18.75%) with stage III tumors than WA-TNBC (11.76%) (*Figure 1F*). Collectively, disparity in incidence, survival, as well as prevalence of higher-grade tumors at the time of diagnosis among AA and WA women indicated toward a biological basis and led us to test the hypothesis that TNBC arising in AA women are inherently wired for aggressive growth and metastatic progression. First, using multiple TNBC cell lines, we compared the growth and proliferation of AA- and WA-TNBC cells. AA-TNBC cells exhibited higher growth ranging from 2.5- to 4-fold (MDA-MB-468 – 2.5-fold, HCC1569 – 3-fold and HCC1806 – 4-fold) within 96 hr compared to WA-TNBC cells which exhibited growth ranging from 1.8- to 2.5-fold (Hs578t – 1.8-fold, HCC1937 – 2-fold, and BT549 – 2.5-fold) (*Figure 1G*). Trypan blue dye exclusion assay also revealed a 2.5-fold increased growth of AA-TNBC cells in comparison to WA-TNBC cells (*Figure 1J*). Next, to examine whether migratory potential of AA-TNBC cells differs from WA-TNBC cells, we performed scratch-migration and spheroid-migration assay. More rapid wound closure within 12 hr was observed for AA-TNBC cells (HCC1806 and HCC1569) in comparison to WA-TNBC cells (Hs578t and HCC1937) in a scratch-migration assay (*Figure 1H*). HCC1806 and HCC1569 migrated at a speed of 3.1 and 2.8 μm/hr whereas Hs578t and HCC1937 migrated at a speed of 1.8 μm/hr (*Figure 1K*). Further validation of our observations using a 3D spheroid-migration assay showed increased migration of AA-TNBC cells from the spheroids (*Figure 1I*). HCC1806 and HCC1569 cells migrated an average distance of 29 and 27.72 μm. In the case of Hs578t and HCC1937 spheroids, the distance traversed was 15.45 and 10.08 μm. AA-TNBC cells exhibited higher migration rate (1.209 and 1.155 μm/hr for HCC1806 and HCC1569 cells) than WA-TNBC cells which migrated at a speed of 0.644 (Hs578t) and 0.42 μm/hr (HCC1937) (*Figure 1L*). These results reveal that AA-TNBC cells have higher migratory potential compared to WA-TNBC cells.

Interestingly, gene set enrichment analysis (GSEA) of the differentially expressed genes (DEGs) in AA-TNBC and WA-TNBC diagnosed between 1987 and 2007 (GSE46581 dataset) (*Lindner et al., 2013*) showed a positive enrichment of epithelial mesenchymal transition pathway and metastasis pathway in AA-TNBC tumors compared to WA-TNBC tumors (*Figure 2A*). Furthermore, analyses of the RNA-sequencing data of TNBC cell lines from Broad Institute Cancer Cell Line Encyclopedia (CCLE) (available with Expression Atlas of EMBL-EBI database) revealed an upregulation of an embryonic stem cell pluripotency pathway in MDA-MB-468, HCC1806, and HCC1569 (AA-TNBC) cells in comparison to HCC1937, BT20, MDA-MB-231, Hs578t, and BT549 (WA-TNBC) cells. Increased expression of a set of 23 embryonic stemness-associated genes was noted in AA-TNBC cells compared to WA-TNBC cell lines (*Figure 2B*). Inspired by the results of the in silico analysis indicating inherently higher stemness in AA-TNBC cells, we assessed the self-renewal potential of AA-TNBC and WA-TNBC cells. Higher number of mammospheres (~2-fold increase) were formed by AA-TNBC cells in contrast to WA-TNBC cells which formed fewer mammospheres. Also, the mammospheres formed by AA-TNBC cells were larger in size in comparison to WA-TNBC mammospheres (*Figure 2C and D*). Next, we examined AA-TNBC and WA-TNBC cells for the presence of stemness markers, CD44 and CD49f, using FACS analysis. Higher population of CD44$^{high}$/CD49f$^{high}$ breast cancer stem

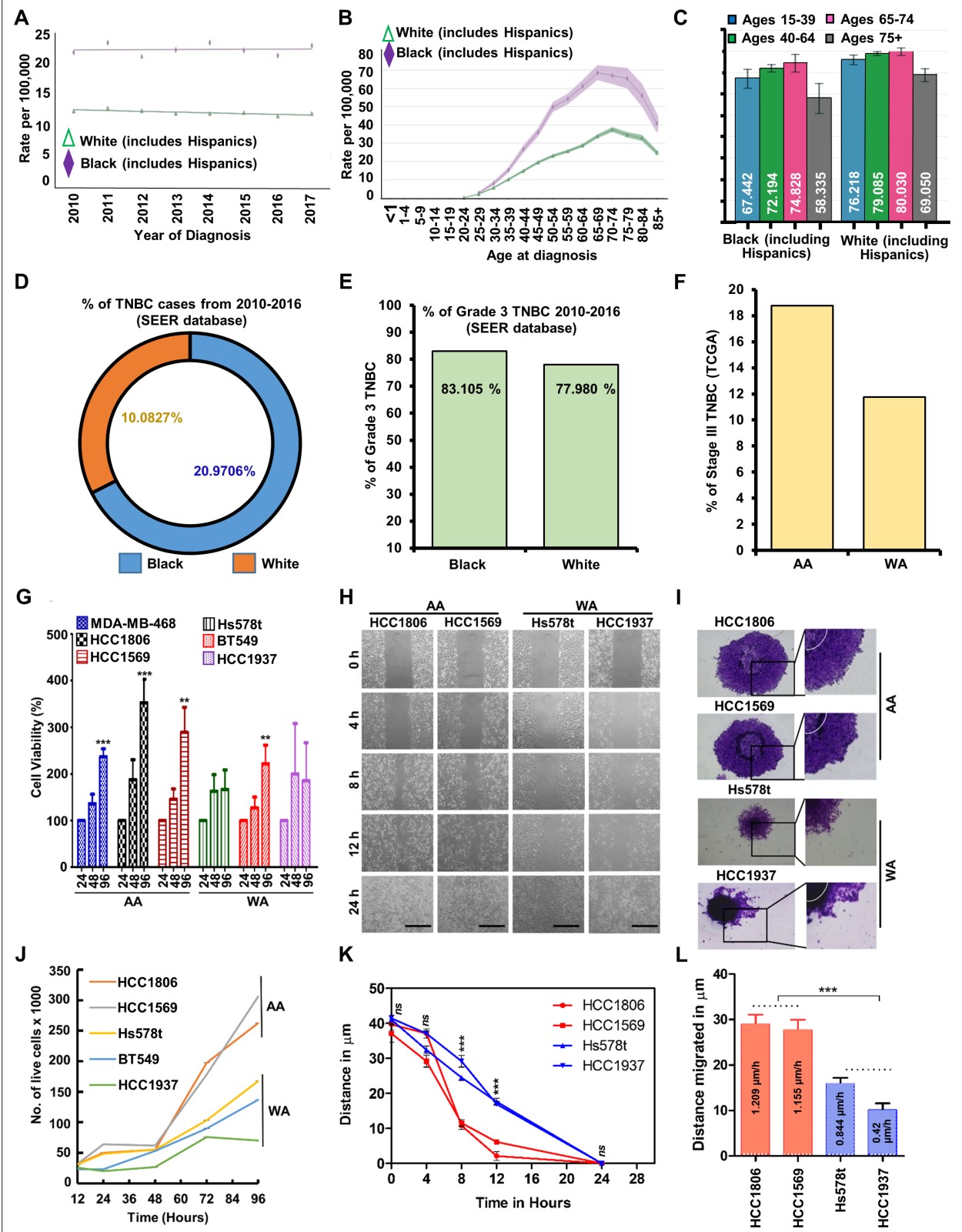

**Figure 1.** African American (AA) women exhibit higher triple negative breast cancer (TNBC) incidence and related mortality and AA-TNBC cells possess higher proliferation and migration potential compared to White American (WA) women. (**A**) Race-specific incidence rate of TNBC (female only) (Surveillance, Epidemiology, and End Results [SEER] data). (**B**) Age-wise incidence of TNBC per 100,000 black (including Hispanics) vs. white (including Hispanics) women. (**C**) Five-year relative survival rates of TNBC (female only) in black (including Hispanics) vs. white women (including Hispanics). (**D**)

*Figure 1 continued on next page*

Figure 1 continued

Percentage of TNBC cases in AA vs. WA women (SEER data). (E) Percentage of Grade 3 tumors in AA and WA women (SEER data). (F) Percentage of Stage III TNBC among AA vs. WA women from the TCGA dataset. (G) Bar graph shows % cell viability of MDA-MB-468, HCC1806, HCC1569, Hs578t, BT549, and HCC1937 cells. (H, K) Representative images of HCC1806, HCC1569, Hs578t, and HCC1937 cells undergoing scratch migration assay. Images are captured at different time intervals as indicated. Graph shows a quantitative representation of distance remaining in the original scratch (width) as cells migrate over time. Scale bar 40 µm. (I, L) Spheroid migration of HCC1806, HCC1569, Hs578t, and HCC1937 cells. Representative images of spheroids are shown. Graph shows average distance migrated. Data represents n = 3 independent experiments. *p ≤ 0.05, **p ≤ 0.01, ***p ≤ 0.001 (J) HCC1806, HCC1569, Hs578t, BT549, and HCC1937 cells were grown for 12, 24, 36, 48, 60, 72, 84, and 96 hr, and then subjected to trypan blue dye exclusion assay.

cell (BCSCs) was observed in AA-TNBC HCC1569 (80.7%) and HCC1806 (93.4%) while MDA-MB-468 cells showed slightly decreased levels (15.7%). Among WA-TNBC cells, Hs578t exhibited much lower levels of CD44$^{high}$/CD49f$^{high}$ BCSCs (1.6%) whereas BT549 and HCC1937 cells showed slightly higher BCSCs (52.5% and 55.4%). Although variation in the CD44$^{high}$/CD49f$^{high}$ population was noted among the AA-TNBC and WA-TNBC cells, AA-TNBC group presented higher overall stemness (*Figure 2E*). In addition, AA-TNBC cells showed elevated expression of embryonic stem cell markers, NANOG, OCT4, SOX2, KLF4, and c-MYC, compared to WA-TNBC cells (*Figure 2F*). Increased nuclear localization of cMyc, Oct4, Nanog, and Sox2 was also observed in AA-TNBC cells in contrast to WA-TNBC cells (*Figure 2G*). Furthermore, analysis of Affymetrix and TCGA datasets revealed higher expression of CD44, KLF4, MYC, and NANOG in TNBC tumors from AA women compared to TNBC tumors from WA women (*Figure 2H*). Collectively, these data show that TNBC in AA women have an inherent propensity for higher growth, migration, and stemness potential.

## GLI1 and Notch1 pathways are inherently upregulated in AA-TNBC cells

Next, we queried the signaling pathways that underlie the racial disparity between AA- and WA-TNBC. To address this question, we further examined the RNA-sequencing data of TNBC cell lines from CCLE (available with Expression Atlas of EMBL-EBI database) encompassing MDA-MB-468, HCC1806, and HCC1569 (AA-TNBC), and HCC1937, BT20, MDA-MB-231, Hs578t, and BT549 (WA-TNBC) cells. DEGs analysis of the global differences in RNA transcript levels in AA-TNBC cell line group vs. WA-TNBC cell line group revealed an increased expression of GLI1 and Notch1 pathways in AA-TNBC. Genes associated with GLI1 (marked in red) and Notch1 (marked in blue) pathways are highlighted in the volcano plot (*Figure 3A*). Genes specifically associated with hedgehog-GLI1 pathway are presented in a heatmap (*Figure 3—figure supplement 1*). Further exploration of GLI1 in AA-TNBC showed elevated expression of GLI1, SHH, and FOXM1 in AA-TNBC cells while minimal expression was observed in WA-TNBC cells (*Figure 3B*). Then we analyzed whether the expression of GLI1 and GLI1-responsive genes correlated with recurrence-free survival in a cohort of TNBC patients. Kaplan–Meier analysis showed that elevated expression of GLI1 and GLI1-responsive genes strongly associated with poor recurrence-free survival in TNBC patients (*Figure 3C*). In addition, higher expression of GLI1-responsive genes, CCNF, FOXM1, GSK3, and CDC34, was observed in TNBC tumors from AA women compared to WA women in Affymetrix and TCGA datasets (*Figure 3D*). GLI1, the effector molecule of oncogenic hedgehog-GLI1 pathway, translocates to nucleus upon activation enabling the transcription of GLI1-responsive genes (*Bhateja et al., 2019*). Immunofluorescence analysis showed elevated nuclear accumulation of GLI1 in AA-TNBC cells compared to WA-TNBC cells (*Figure 3E and F*). Higher levels of nuclear GLI1 was observed in nuclear extracts of AA-TNBC cells while WA-TNBC cells exhibited very low levels of GLI1 (*Figure 3G*). These results explicitly show that AA-TNBC cells harbor elevated levels of GLI1.

Since we noted an enrichment of Notch1 pathway genes in the RNA-sequencing of AA-TNBC cell line group compared to WA-TNBC cell line group, we next examined the involvement of Notch1 in AA-TNBC. GSEA of the DEGs in AA-TNBC and WA-TNBC (GSE46581 dataset) (*Lindner et al., 2013*) showed a positive enrichment of Notch1 pathway in AA-TNBC tumors in comparison to WA-TNBC tumors (*Figure 4A*). Expression levels of Notch1 pathway genes are presented in a heatmap (*Figure 4—figure supplement 1*). Analysis of the TCGA dataset showed higher Notch1 expression in AA-TNBC compared to WA-TNBC samples (*Figure 4B*). Next, we analyzed the association between Notch1 expression and survival of TNBC patients by Kaplan–Meier survival analysis.

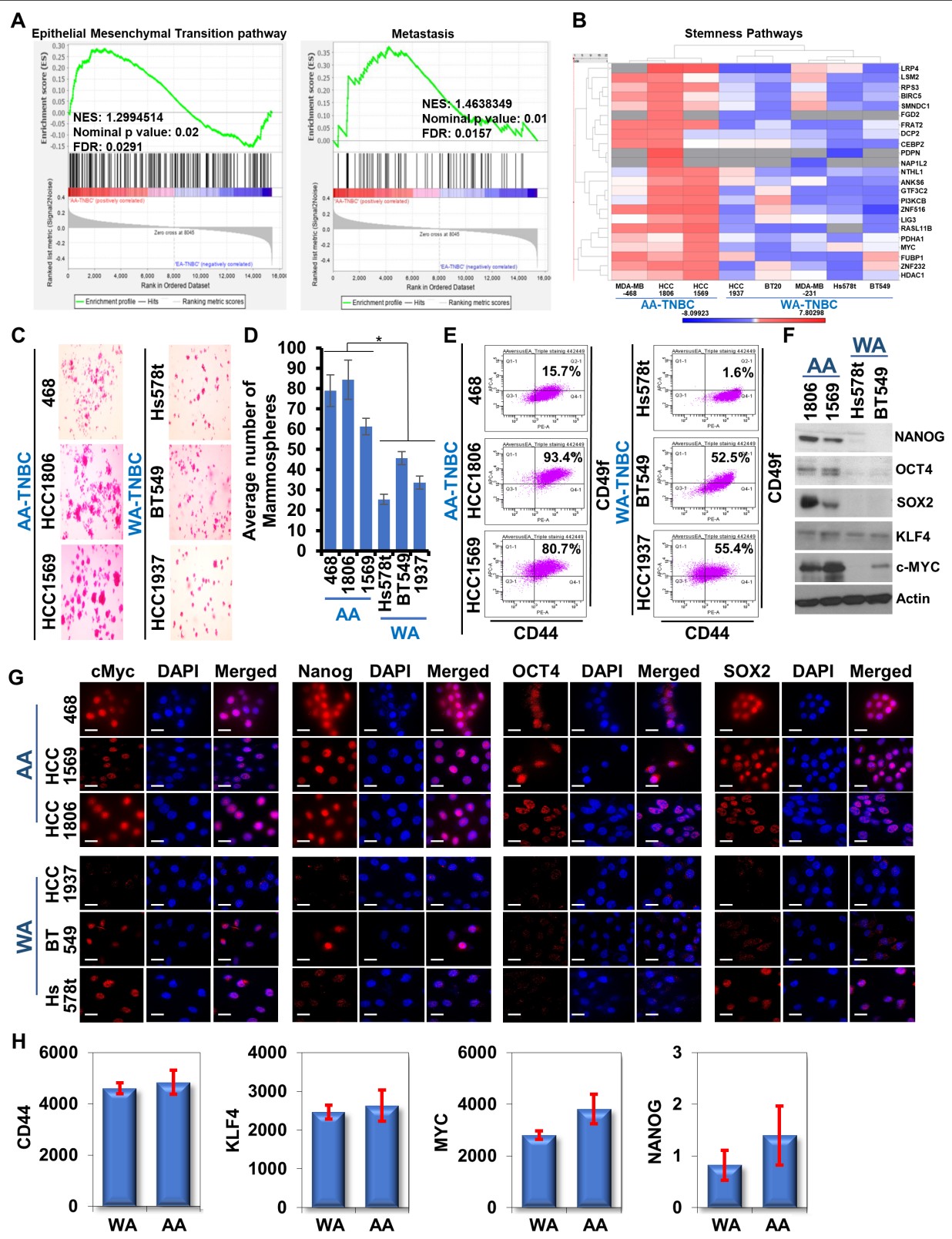

**Figure 2.** African American (AA)-triple negative breast cancer (TNBC) cells possess higher stem cell enriched population than White American (WA)-TNBC cells. (**A**) Gene set enrichment analysis (GSEA) of hallmark epithelial-mesenchymal transition and Tavazoie metastasis pathway in cancer between AA-TNBC samples and WA-TNBC samples from GSE46581 dataset (Gene Expression Omnibus). (**B**) Differential expression statistics comparing AA-TNBC cells (MDA-MB-468, HCC1806, HCC1569) with WA-TNBC cells (Hs578t, BT549, BT20, HCC1937, MDA-MB-231) were calculated and results are

*Figure 2 continued on next page*

*Figure 2 continued*

shown in a heatmap (p ≤ 0.05). Genes are marked on the side. (**C, D**) Representative images of mammospheres formed by HCC1806, HCC1569, MDA-MB-468, HCC1937, BT549, and Hs578t cells. Bar graphs show quantification of solid mammospheres. Data represents n = 4 independent experiments. *p ≤ 0.05. (**E**) Flow cytometry analysis of AA-TNBC and WA-TNBC cells using CD44 and CD49f staining. (**F**) Immunoblot analysis of NANOG, OCT4, SOX2, KLF4, and c-MYC in HCC1806, HCC1569, Hs578t, and BT549 cells. Expression of ACTB served as the loading control. (**G**) Immunofluorescence analysis of cMyc, OCT4, NANOG, and SOX2 in MDA-MB-468, HCC1569, HCC1806, HCC1937, BT549, and Hs578t cells. Nuclei are stained with DAPI. Scale bar, 20 µm. (**H**) Normalized mRNA expression of CD44 (*p = 0.102), KLF4 (*p = 0.010), MYC (*p = 0.002), and NANOG (*p = 0.023) in WA- and AA-TNBC patients from TCGA and Affymatrix datasets.

The online version of this article includes the following figure supplement(s) for figure 2:

**Source data 1.** Western blot data for *Figure 2F*.

Strong association between high Notch1 expression and poor RFS was observed with hazard ratio of 1.97 (p = 0.0021) (*Figure 4C*). Membrane-bound Notch receptor undergoes cleavage upon activation and releases the cytoplasmic domain or Notch intracellular domain (NICD), which translocates to the nucleus and mediates transcriptional activation of Notch-responsive genes (*Brzozowa-Zasada et al., 2017*). Immunoblot analysis of TNBC cells exhibited elevated expression of NICD, JAGGED, and HES1 in AA-TNBC cells whereas WA-TNBC cells showed low expression levels (*Figure 4D*). Further exploration of Notch1 signaling in Affymetrix and TCGA datasets showed higher expression of Notch1-responsive genes, HEY1, HEY2, HES1, HES4, HES5, and HES6, in AA-TNBC compared to WA-TNBC tumors (*Figure 4E*). As nuclear translocation of NICD is the hallmark of induced Notch signaling cascade (*Brzozowa-Zasada et al., 2017*), we analyzed the nuclear localization of Notch1 in multiple AA-TNBC and WA-TNBC cells. Confocal microscopy detected higher nuclear accumulation of Notch1 in AA-TNBC compared to WA-TNBC cells (*Figure 4F*). We observed more than 4-fold higher nuclear accumulation of Notch1 in AA-TNBC cells compared to WA-TNBC cells (*Figure 4G*). Next, nuclear and cytoplasmic lysates from AA-TNBC and WA-TNBC cells were queried for the level of NICD. Elevated levels of nuclear Notch1 was noted in AA-TNBC cells while minimal expression of NICD was observed in nuclear extracts of WA-TNBC cells. Very low to no expression of NICD was observed in cytoplasmic extracts of AA as well as WA-TNBC cells (*Figure 4H*). These data clearly present that AA-TNBC cells possess inherently higher levels of NICD.

## GLI1 interacts with Notch1 in AA-TNBC and their combined inhibition synergistically inhibits AA-TNBC

Having observed an elevated expression of GLI1 and Notch1 in AA-TNBC, we hypothesized that GLI1 and NICD may exhibit co-localization and direct interaction in AA-TNBC cells. To address this hypothesis, a co-expression analysis of GLI1 gene signature vs. Notch1 gene signature was conducted in AA-TNBC vs. WA-TNBC samples from TCGA dataset. Single sample gene set enrichment analysis (ssGSEA) revealed a positive correlation between GLI1-associated genes signature and Notch1-associated gene signature in AA-TNBC tumors (n = 32, R = 0.4845) while a negative correlation was found in WA-TNBC patients (n = 68, R = –0.2220) (*Figure 5A*). However, these correlations are not strong and adding more clinical samples for AA-TNBC and WA-TNBC can likely strengthen these observations. We examined the physical interaction between GLI1 and NICD in AA-TNBC and WA-TNBC cells using immunoprecipitation and immunofluorescence analysis. Flag-tagged GLI1 was expressed in HCC1569 cells using hgliflag3x plasmid. Immunoprecipitates with GLI1 antibody showed the presence of NICD while IgG controls were negative (*Figure 5B*). We further evaluated the interaction between nuclear GLI1 and nuclear NICD endogenously present in AA-TNBC and WA-TNBC cells. NICD was immunoprecipitated with GLI1 antibody in the nuclear extracts of AA-TNBC cells while low/no presence of NICD was observed in WA-TNBC cells (*Figure 5C*). In a reciprocal analysis, GLI1 was immunoprecipitated with NICD antibody in the nuclear extracts of AA-TNBC cells while low/no presence of GLI1 was observed in WA-TNBC cells (*Figure 5C*). Next, we investigated whether GLI1 overexpression influences transcriptional activity of NICD in AA-TNBC. Cells overexpressing GLI1 exhibited increased NICD transactivation in a luciferase assay (*Figure 5D*) and also showed increased expression of Notch-responsive genes – Hes1 and Hey1 (*Figure 5E*). Notch1 overexpression also led to increased expression of GLI1-responsive gene – FOXM1 (*Figure 5E*). These findings were also corroborated with immunofluorescence analysis where increased co-localization of GLI1 and NICD was observed in AA-TNBC cells in comparison to WA-TNBC cells which exhibited very low expression

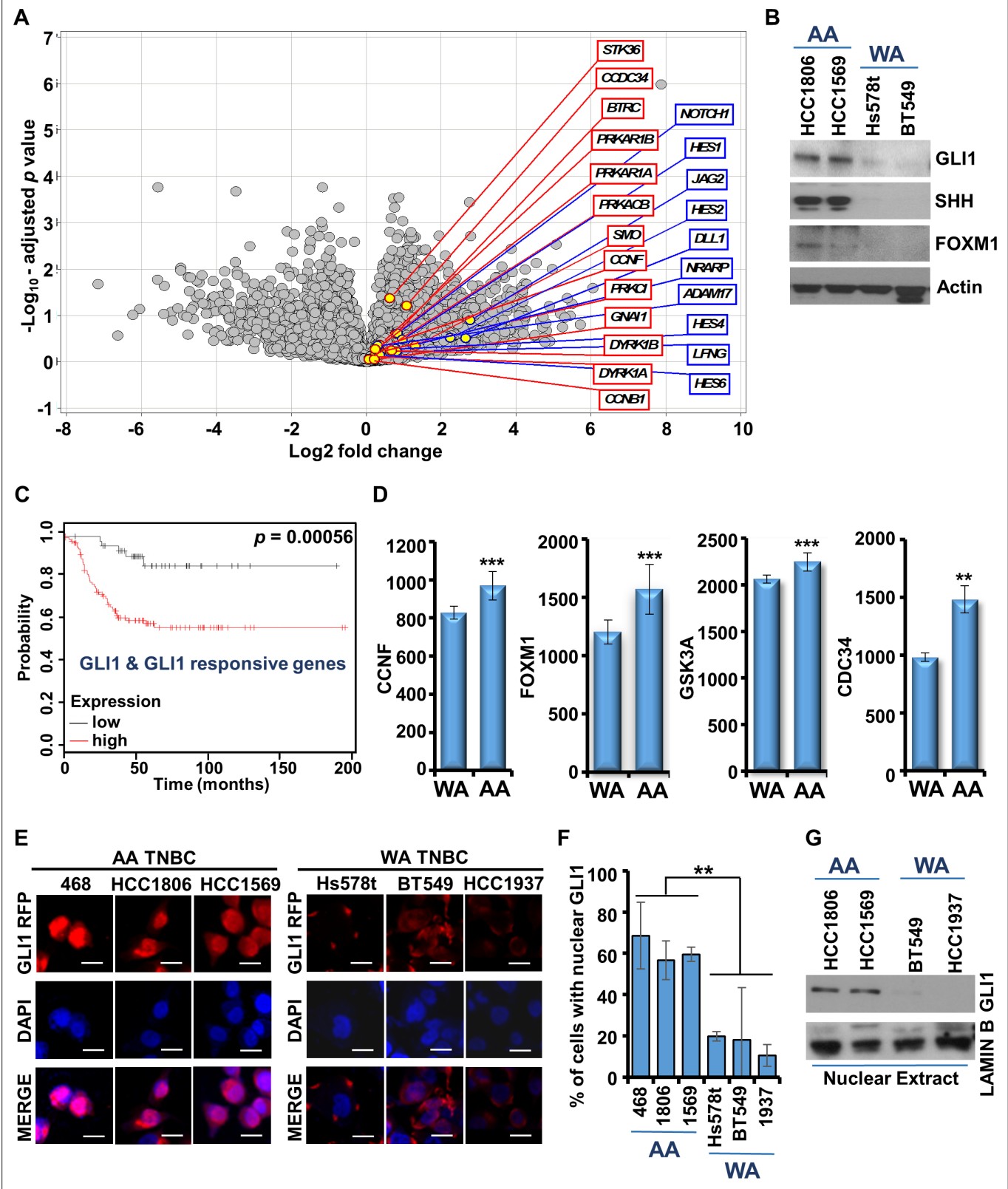

**Figure 3.** GLI1 and Notch1 pathways are upregulated in African American (AA)-triple negative breast cancer (TNBC) in comparison to White American (WA)-TNBC. (**A**) Differential expression statistics comparing AA-TNBC cells with WA-TNBC cells were calculated and results are shown in a volcano plot. Upregulation of GLI1-responsive genes (marked in red) and Notch1 responsive genes (marked in blue) as observed in gene expression analyses is shown. (**B**) Immunoblot analysis of GLI1, SHH, and FOXM1 in HCC1806, HCC1569, Hs578t, and BT549 cells. Expression of actin serves as the loading

*Figure 3 continued on next page*

Figure 3 continued

control. (C) Kaplan–Meier curve showing recurrence-free survival (RFS) for the TNBC patients with high or low GLI1 and GLI1 target gene expression. *p = 0.00056. (D) Normalized mRNA expression of GLI1 target genes in WA-TNBC and AA-TNBC patients, respectively, from TCGA cohort. **p ≤ 0.01, ***p ≤ 0.001. (E, F) Immunofluorescence analysis of GLI1 in MDA-MB-468, HCC1806, HCC1569, Hs578t, BT549, and HCC1937 cells. Bar graphs show quantitation of cells with nuclear GLI1 expression. Data represents n = 4 independent experiments. **p ≤ 0.01. Scale bar, 20 µm. (G) Immunoblot analysis of GLI1 in the nuclear extracts in HCC1806, HCC1569, BT549, and HCC1937 cells. Lamin B is used as control.

The online version of this article includes the following figure supplement(s) for figure 3:

**Source data 1.** Western blot data for *Figure 3B*.

**Source data 2.** Western blot data for *Figure 3G*.

**Figure supplement 1.** Increased expression of GLI1 pathway genes in African American (AA)-triple negative breast cancer (TNBC) cell lines compared to White American (WA)-TNBC cell lines.

of GLI1 and NICD (*Figure 5F*). These results show that GLI1 and NICD, effector transcription molecules of oncogenic hedgehog-GLI and Notch pathways, interact in AA-TNBC cells.

Having observed inherently high levels of GLI1 and Notch1 along with their interaction in AA-TNBC, we hypothesized that combined inhibition of GLI1 and Notch1 might prove valuable to inhibit AA-TNBC. Before evaluating the combinational effect of GLI inhibitor, GANT61 and Notch inhibitor, DAPT, we examined their ability to inhibit AA-TNBC cells as single agents. Treatment with 20 µM GANT61 led to 50% growth inhibition but no $IC_{50}$ was achieved in the DAPT-treated AA-TNBC cells (*Figure 5—figure supplement 1A,C,E*). Next, AA-TNBC cells were treated with various combinations of DAPT and GANT61 to find whether the combination of GANT61 and DAPT is additive, synergistic, or antagonistic. Analysis of the viability data using Compusyn software (Compusyn Inc,. Pramus, NJ) uncovered the synergistic effects of combination regimens (*Figure 5—figure supplement 1B,D,F*). Next, we examined the effect of DAPT and GANT61 on clonogenicity and observed that combined treatment with DAPT and GANT61 effectively inhibited clonogenicity of MDA-MB-468, HCC1806, and HCC1569 cells compared to single agents (*Figure 5E*). Conversely, WA-TNBC cells (BT549, MDA-MB-231, and HCC1937) did not show any inhibition of clonogenicity in response to single/combination regimen of DAPT and GANT61 (*Figure 5E*). Similar results were observed in cell viability assay where WA-TNBC cells showed no significant inhibition upon DAPT/GANT61 treatment while AA-TNBC cells were significantly inhibited (*Figure 5—figure supplement 2*). Owing to the lack of tangible markers and ensuing targeted therapies, TNBC tumors are primarily treated with chemotherapy regimens. We questioned whether inhibiting GLI1 and Notch1 would improve the efficacy of doxorubicin and carboplatin in AA-TNBC. HCC1806 cells treated with 50 µM DAPT, 10 µM GANT61 in combination with a varied concentration of doxorubicin exhibited a 10-fold reduced $IC_{50}$ in DOX + GANT61+ DAPT combination compared to doxorubicin monotherapy (*Figure 5F*). While cells treated with 40 µM of carboplatin monotherapy did not achieve $IC_{50}$, surprisingly, treatment with 0.25 µM of carboplatin in combination with 50 µM DAPT and 10 µM GANT61 caused 50% cell death in HCC1806 cells (*Figure 5G*). Dose-effect analysis of the combination regimens (DAPT + GANT61 + DOX/CARBO) compared to either monotherapy (DOX/CARBO) revealed significant synergistic interactions indicating the superior efficacy of combination regimens (*Figure 5H–K*). Together, these data show that AA-TNBC cells exhibit an interaction between GLI1 and NICD and combined inhibition of GLI1 and Notch1 along with chemotherapy can potentially improve the efficacy of chemotherapy regimens.

## Triple-combination therapy effectively inhibits tumor growth and stemness in AA-TNBC

Rapid growth of AA-TNBC tumors and their propensity to relapse with distant metastases emphasize the dire need for the development of new therapeutic regimens based on their unique biology. First, we investigated the effectiveness of combination treatment with GLI1 and Notch1 inhibitors in comparison to doxorubicin and carboplatin combination in AA-TNBC and WA-TNBC tumors. Mice harboring HCC1806-, HCC1937-, and MDA-MB-231-derived tumors were regularly treated with DAPT + GANT61 combination or doxorubicin + carboplatin combination and tumor progression was monitored. AA-TNBC (HCC1806) tumors showed more effective growth inhibition in response to DAPT + GANT61 in comparison to chemotherapy combination whereas WA-TNBC (HCC1937 and

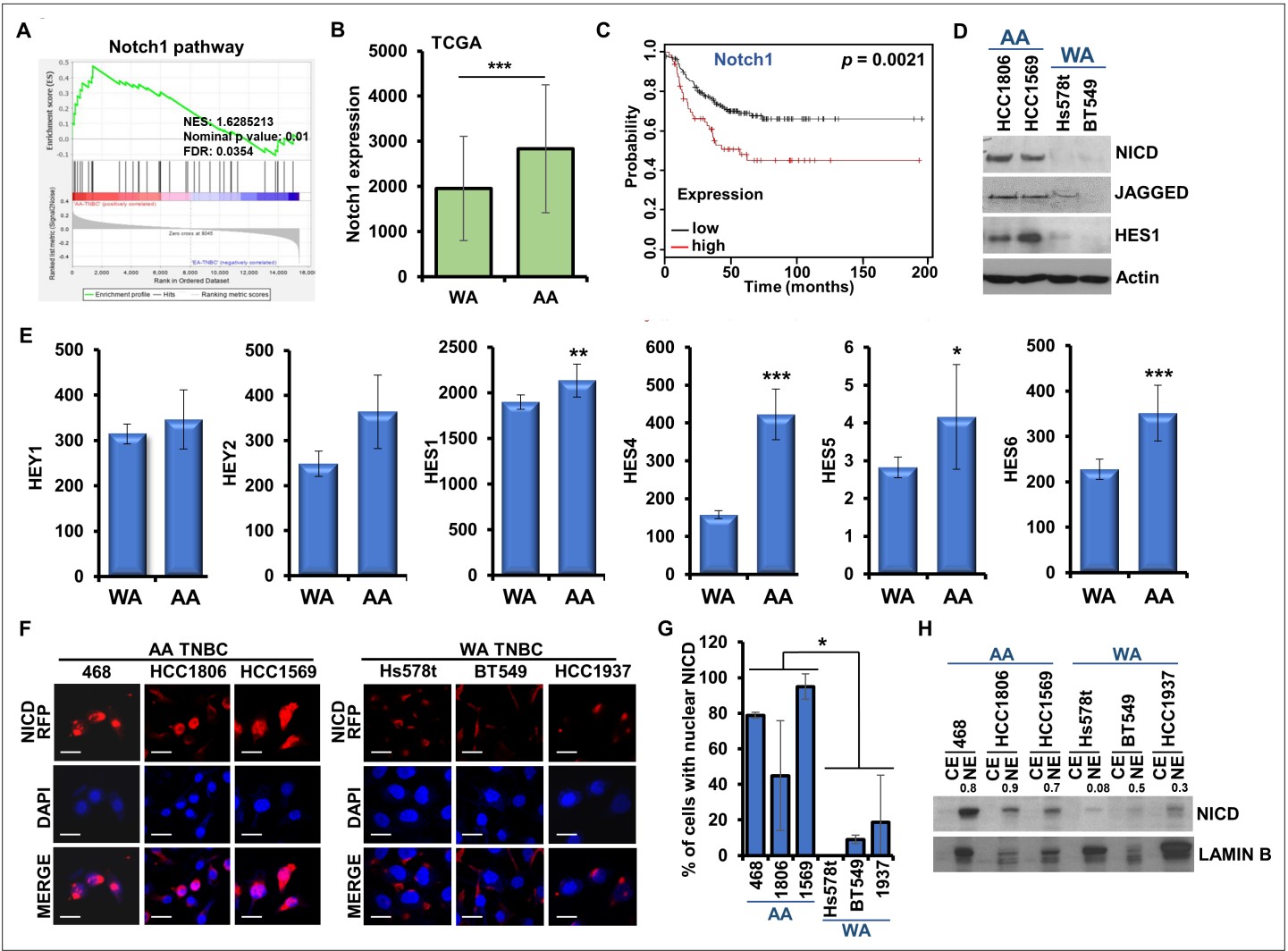

**Figure 4.** African American (AA)-triple negative breast cancer (TNBC) cells exhibit increased activation of Notch1, a key protein associated with poor survival. (**A**) Gene set enrichment analysis (GSEA) of Notch1 pathway between AA-TNBC samples and White American (WA)-TNBC samples from GSE46581 dataset (Gene Expression Omnibus). (**B**) Normalized average Notch1 mRNA expression in AA (n = 32) and WA (n = 68) TNBC patients, respectively, from TCGA cohort. (**p ≤ 0.01). (**C**) Kaplan–Meier curve showing recurrence-free survival (RFS) for the TNBC patients with high and low Notch1 expression. p = 0.0021. (**D**) Immunoblot analysis of Notch intracellular domain (NICD), JAGGED, and HES1 in HCC1806, HCC1569, Hs578t, and BT549. Actin serves as the loading control. (**E**) Normalized mRNA expression of Notch1 target genes (HEY/HES family genes) in WA- and AA-TNBC patients, respectively, from TCGA cohort. (*p ≤ 0.05, **p ≤ 0.01, ***p ≤ 0.001). (**F, G**) Immunofluorescence analysis of NICD in MDA-MB-468, HCC1806, HCC1569, Hs578t, BT549, and HCC1937 cells. Bar graphs show quantitation of cells with nuclear NICD expression. Data represents n = 4 independent experiments. *p ≤ 0.05. Scale bar, 20 µm. (**H**) Immunoblot analysis of NICD in the nuclear extracts in MDA-MB-468, HCC1806, HCC1569, BT549, and HCC1937 cells. Lamin B is used as control.

The online version of this article includes the following figure supplement(s) for figure 4:

**Source data 1.** Western blot data for *Figure 4D*.

**Source data 2.** Western blot data for *Figure 4H*.

**Figure supplement 1.** Upregulated expression of Notch1 pathway genes in African American (AA)-triple negative breast cancer (TNBC) cell lines compared to White American (WA)-TNBC cell lines.

MDA-MB-231) tumors showed enhanced growth inhibition in response to doxorubicin + carboplatin combination. GANT61 + DAPT combination treatment did not reduce the growth of WA-TNBC cells-derived tumors (*Figure 6—figure supplement 1*). Tumor-dissociated cells from WA-TNBC-HCC1937-derived tumors showed no/low reduction of the aldehyde dehydrogenase activity in response to DAPT + GANT61 combination treatment (32.7%) in comparison to vehicle-treated group (36.5%)

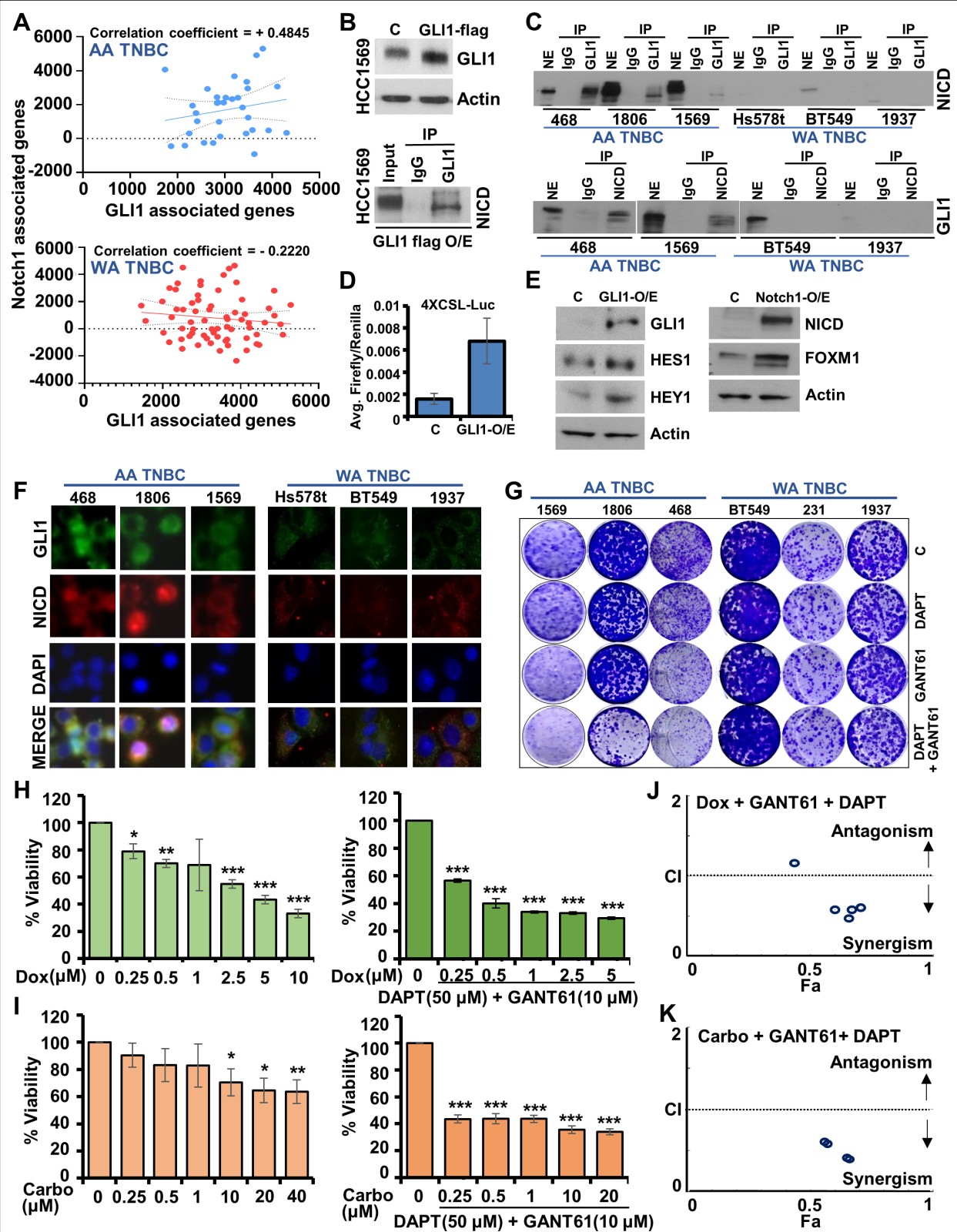

**Figure 5.** Concomitant upregulation, co-localization, and interaction of GLI1 and Notch intracellular domain (NICD) in African American (AA)-triple negative breast cancer (TNBC). (**A**) Co-expression of GLI1 gene signature and Notch1 gene signature in AA-TNBC (n = 32, R = +0.4845) and White American (WA)-TNBC from TCGA cohort (n = 68, R = –0.2220). (**B**) Immunoblot analysis of GLI1 and actin in the nuclear extract of HCC1569 cells transfected with Flag-GLI1 overexpression constructs. Immunoprecipitation of GLI1 from the nuclear extracts of cells transfected with Flag-GLI1

*Figure 5 continued*

overexpression constructs and immunoblotting with NICD antibodies. (**C**) Co-immunoprecipitation of NICD with GLI1 from the nuclear extracts of MDA-MB-468, HCC1806, HCC1569, Hs578t, BT549, and HCC1937 cells. Lower panel shows co-immunoprecipitation of GLI1 with NICD from the nuclear extracts of MDA-MB-468, HCC1569, BT549, and HCC1937 cells. NE (nuclear extract input), IP (immunoprecipitation). (**D**) Luciferase activity of 4XCSL-Luc in MDA-MB-468 cells transfected with GLI1 overexpression construct. (**E**) HCC1806 cells were transfected with Flag-GLI1 overexpression construct and lysates were immunoblotted for Notch-responsive genes – Hey1 and Hes1. HCC1806 cells were transfected with Notch1 overexpression construct and lysates were immunoblotted for GLI1-responsive gene – FOXM1. Actin serves as the loading control. (**F**) Immunofluorescence analysis of NICD and GLI1 in MDA-MB-468, HCC1806, HCC1569, Hs578t, BT549, and HCC1937 cells. Nuclei are stained with DAPI. Scale bar, 20 µm. (**G**) Representative images of colonies formed by AA-TNBC and WA-TNBC cells treated with DAPT (50 µM), GANT61 (10 µM), and the combination of DAPT (50 µM) and GANT61 (10 µM) in a clonogenicity assay. (**H, I**) Cell viability assay of HCC1806 cells treated with doxorubicin, carboplatin, doxorubicin + DAPT (50 µM)+ GANT61 (10 µM) and carboplatin + DAPT (50 µM)+ GANT61 (10 µM) as indicated for 24 hr. *$p \leq 0.05$, **$p \leq 0.01$, ***$p \leq 0.001$. (**J, K**) Synergistic interaction between the fixed concentration of DAPT (50 µM) + GANT61 (10 µM) treatment with a varied concentration of doxorubicin (0.25, 0.5, 1, 2.5, and 5 µM) and carboplatin (0.25, 0.5, 1, 10, and 20 µM), respectively.

The online version of this article includes the following figure supplement(s) for figure 5:

**Source data 1.** Western blot data for *Figure 5B*.

**Source data 2.** Western blot data for *Figure 5B*.

**Source data 3.** Western blot data for *Figure 5C*.

**Source data 4.** Western blot data for *Figure 5E*.

**Figure supplement 1.** Synergistic effect of GLI1 and Notch1 inhibitors in reducing the viability of African American (AA)-triple negative breast cancer (TNBC) cells.

**Figure supplement 2.** Effect of GLI1 and Notch1 inhibitors on the viability of African American (AA)-triple negative breast cancer (TNBC) and White American (WA)-TNBC cells.

and DOX-CARBO group (31.9%). In contrast, AA-TNBC-HCC1806-derived tumors exhibited reduced aldehyde dehydrogenase activity in DAPT + GANT61 treatment group (53.3%) compared to vehicle-treated group (77.5%) and DOX-CARBO (66.4%) (*Figure 6A*, *Figure 6—figure supplement 2*).

Next, encouraged by our in vitro and in vivo findings, we investigated whether a combination of GLI1 and Notch1 inhibitors with chemotherapy would prove effective against AA-TNBC tumors. Mice harboring HCC1806-derived tumors were regularly treated with various single agents as well as triple-combination therapies and tumor progression was monitored for 4 weeks. More rapid tumor progression was observed in control group compared to treatment groups. Corroborating our in vitro findings, mice treated with triple-combination therapies (DAPT + GANT61 + DOX/CARBO) exhibited significantly reduced tumor growth in comparison to mice treated with single agents (*Figure 6B*). Immunohistochemical analyses showed reduced Ki67, Oct4, and Sox2 staining in DAPT + GANT61 + DOX/CARBO-treated tumors in comparison to tumors from DOX-, CARBO-, GANT61-, and DAPT-treated groups, which showed denser staining (*Figure 6C and J*, *Figure 6—figure supplement 3*, *Figure 6—figure supplement 4*). Tumors from different treatment groups were dissociated into single cells and examined for their functional attributes pertaining to migration, invasion, and mammosphere-formation potential. Interestingly, tumor cells from triple-therapy treated groups exhibited significantly reduced transwell migration (*Figure 6D*, *Figure 6—figure supplement 5*) and Matrigel invasion (*Figure 6E*, *Figure 6—figure supplement 6*) compared to vehicle and single agent-treated groups. As expected, reduced migration of tumor cells was observed in tumor cell spheroids formed with tumor cells from triple-therapy-treated groups in comparison to vehicle and single agent-treated groups (*Figure 6F*). Mammosphere formation potential was evaluated using serial dilution of tumor-derived cells. Decreased mammosphere formation was observed at all dilutions (500–8000 cells/well) for tumor cells from triple-therapy-treated group compared to vehicle and single agent-treated groups (*Figure 6G*). Compared to vehicle and single agent treatment groups, reduced mammospheres were formed in DAPT + GANT61 + DOX/CARBO group (*Figure 6H*). Tumor-dissociated cells, examined for the presence of stemness marker CD44$^+$/CD24$^-$, showed higher stemness in vehicle-treated group (87.6%) compared to triple-therapy-treated tumors (48.6% for DGD and 65.2% in CGD) (*Figure 6I*). Further, immunohistochemical analyses showed higher staining with OCT4 and SOX2 in vehicle and single agent treatment groups compared to DAPT + GANT61 + DOX/CARBO-treated tumors, which showed reduced OCT4/SOX2 staining indicating reduced stemness (*Figure 6J*, *Figure 6—figure supplement 4*).

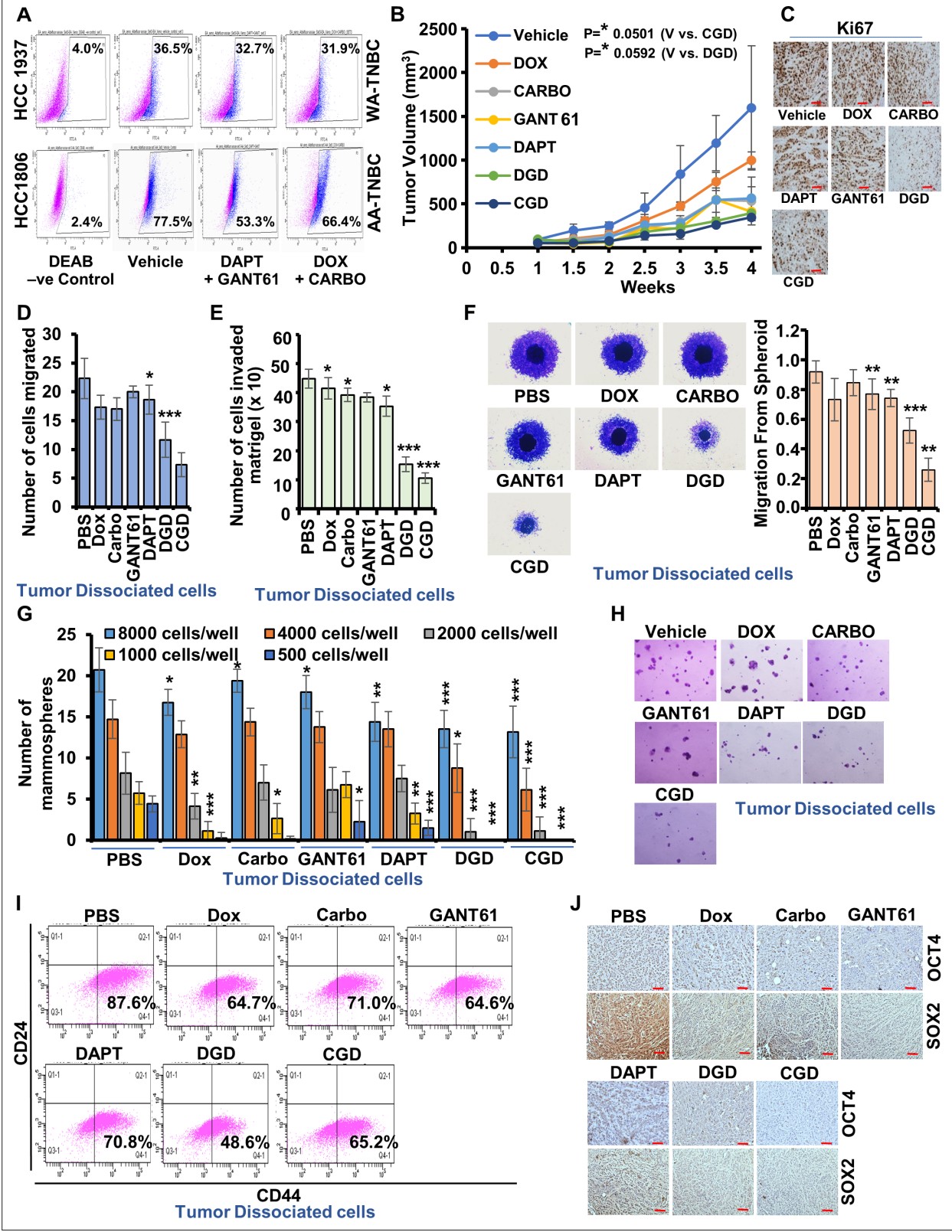

**Figure 6.** Combined treatment with chemotherapy, Notch inhibitor, and GLI1 inhibitor blocks African American (AA)-triple negative breast cancer (TNBC) tumor progression. (**A**) ALDH activity assay for dissociated tumor cells from HCC1937 and HCC1806 tumors treated with DAPT + GANT61 and doxorubicin + carboplatin combination. (**B**) Line graph shows tumor progression curves of HCC1806-derived tumors (n = 6) treated with vehicle, doxorubicin, carboplatin, GANT61, DAPT, DAPT + GANT61 + doxorubicin and CARBO + GANT61 + DAPTplatin. (**C**) Representative images of Ki67

*Figure 6 continued on next page*

*Figure 6 continued*

stained tumor sections from HCC1806-derived tumors. Scale bar, 100 µm. (**D–H**) Resected tumors from control and treated groups were dissociated and isolated tumor cells were subjected to in vitro functional assays. (**D**) Bar graph shows the migration potential of tumor cells isolated from various treatment groups and control. (**E**) Bar graph shows the number of tumor cells invaded through Matrigel. (**F**) Tumor cell spheroids were formed and allowed to migrate. Representative images of spheroids are shown. Graph shows average distance migrated. Data represents n = 3 independent experiments. *p ≤ 0.05, **p ≤ 0.01, ***p ≤ 0.001. (**G**) Bar graph shows quantification of solid mammospheres formed from various numbers of tumor cells isolated from various treatment groups and control. Data represents n = 3 independent experiments. *p ≤ 0.05, **p ≤ 0.01, ***p ≤ 0.001. (**H**) Representative images of mammospheres formed from tumor-dissociated cells. (**I**) Flow cytometry analysis of dissociated tumor cells using CD44 and CD24 staining. (**J**) Representative images of Sox2 and Oct4 stained tumor sections from HCC1806-derived tumors from control and treatment groups. Scale bar, 100 µm.

The online version of this article includes the following figure supplement(s) for figure 6:

**Figure supplement 1.** Effect of GLI inhibitor + Notch inhibitor combination and doxorubicin + carboplatin combination on African American (AA)-triple negative breast cancer (TNBC) and White American (WA)-TNBC cells.

**Figure supplement 2.** African American (AA)-triple negative breast cancer (TNBC) tumors treated with a combination of Notch inhibitors and GLI inhibitor exhibit reduced ALDH activity.

**Figure supplement 3.** African American (AA)-triple negative breast cancer (TNBC) tumors treated with a combination of chemotherapy, Notch inhibitor, and GLI1 inhibitor exhibit reduced Ki-67 expression.

**Figure supplement 4.** African American (AA)-triple negative breast cancer (TNBC) tumors treated with a combination of chemotherapy, Notch inhibitor, and GLI1 inhibitor exhibit reduced Oct4 and Sox2 expression.

**Figure supplement 5.** Combined treatment with chemotherapy, Notch inhibitor, and GLI1 inhibitor reduces migration potential of African American (AA)-triple negative breast cancer (TNBC) tumor-derived cells.

**Figure supplement 6.** Combined treatment with chemotherapy, Notch inhibitor, and GLI1 inhibitor reduces invasion potential of African American (AA)-triple negative breast cancer (TNBC) tumor-derived cells.

AA-TNBC cells inherently harbor higher stemness potential. Tumor-dissociated cells from CARBO + GANT61 + DAPT group consistently showed decreased migration, invasion potential, and mammosphere formation in comparison to CARBO and vehicle-treated groups prompting us to investigate if triple-therapy reduced stemness of AA-TNBC tumors. HCC1806-derived tumors from mice treated with vehicle, CARBO, and CARBO + GANT61 + DAPT groups were excised and dissociated to single cells followed by secondary transplants in limiting dilution (*Figure 7A*). Regular monitoring of tumor incidence and progression uncovered that secondary xenografts from vehicle and CARBO-treated group showed larger tumors and a shorter tumor-free survival while the CARBO + GANT61 + DAPT-treated group exhibited smaller tumor load as well as a longer tumor-free survival (*Figure 7B and C*). CARBO + GANT61 + DAPT-treated tumor group exhibited significantly decreased tumor-initiating frequencies when transplanted into secondary hosts at limiting dilutions. Stem cell frequency was calculated based on tumor incidence in all treatment groups using L-Calc software for limiting dilution analysis (LDA). The frequency of breast tumor-initiating cells in the CARBO + GANT61 + DAPT-treated tumor group (CGD) was determined to be 1 in 1,178,112 cells compared to 1 in 70,160 cells in the control group and CARBO-treated group at week 3. Analysis at week 4 showed that the frequency of breast tumor-initiating cells in the CGD group remained at 1 in 1,178,112 cells in comparison to 1 in 6256 cells in the control group and CARBO-treated group. Of interest, CGD group consistently showed diminished stem cell frequency compared to CARBO-treated/vehicle group (*Figure 7D*). Immunohistochemical analyses of tumors from CARBO + GANT61 + DAPT-treated group exhibited reduced levels of NICD and GLI1 expression in comparison to vehicle and CARBO-treated groups (*Figure 7—figure supplement 1*). Tumor-dissociated cells from CARBO + GANT61 + DAPT-treated group exhibited reduced aldehyde dehydrogenase activity (17.7%) in comparison to vehicle (45.5%) and CARBO (40.3%) (*Figure 7E*). Also, reduced percentage of ALDH+ cells were observed in CARBO + GANT61 + DAPT-treated group (2.5%) compared to vehicle (14.4%) and CARBO (10.4%) (*Figure 7F*). CARBO + GANT61 + DAPT treatment also resulted in reduced percentage of CD44+/CD24- and CD44+/CD49f+ cells in AA-TNBC tumors compared to vehicle and CARBO-treated group (*Figure 7—figure supplement 2*). Together, these results unequivocally present that triple-combination therapy effectively inhibits the growth of AA-TNBC tumors and abrogates the highly migratory, invasive, and stemness-rich phenotype of AA-TNBC tumors.

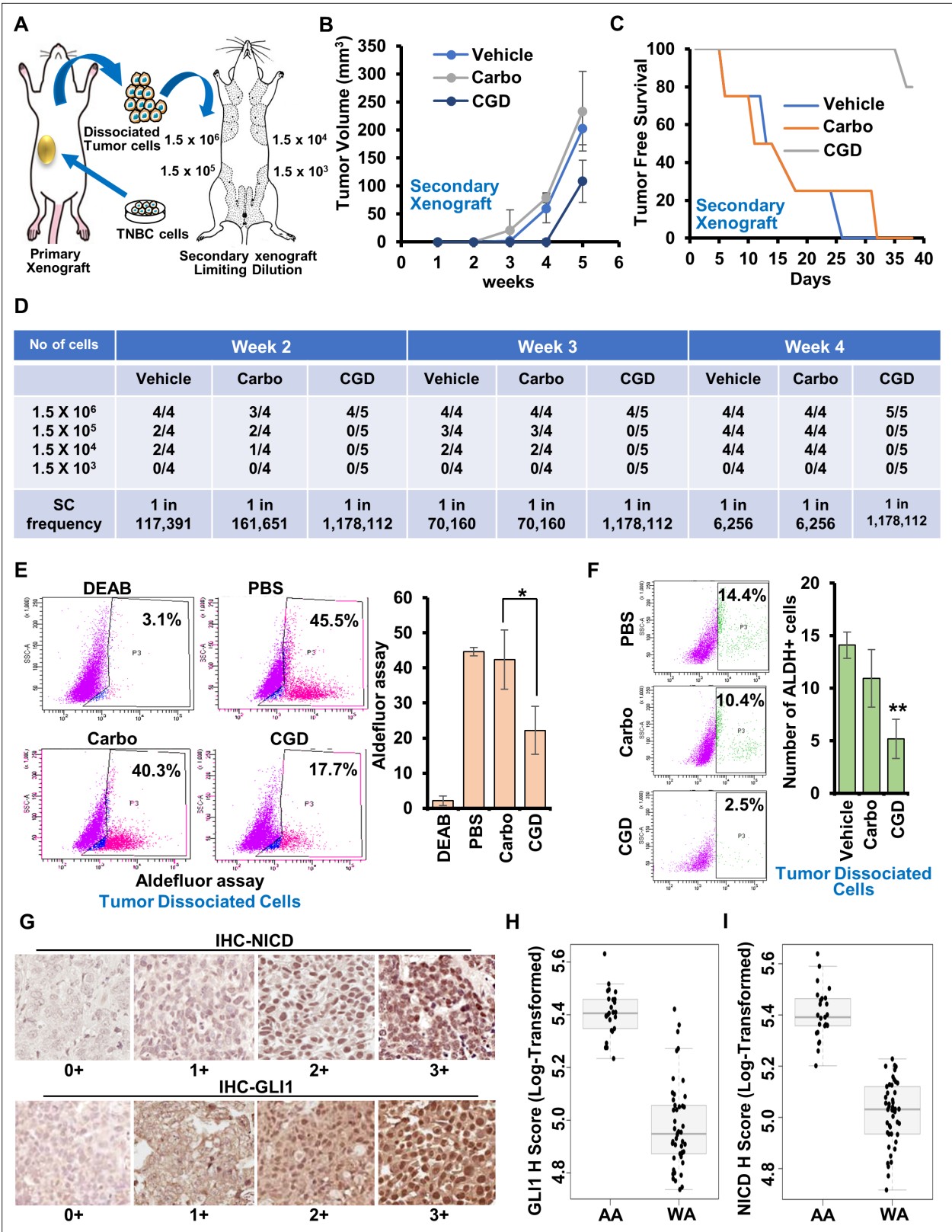

| No of cells | Week 2 | | | Week 3 | | | Week 4 | | |
|---|---|---|---|---|---|---|---|---|---|
| | Vehicle | Carbo | CGD | Vehicle | Carbo | CGD | Vehicle | Carbo | CGD |
| $1.5 \times 10^6$ | 4/4 | 3/4 | 4/5 | 4/4 | 4/4 | 4/5 | 4/4 | 4/4 | 5/5 |
| $1.5 \times 10^5$ | 2/4 | 2/4 | 0/5 | 3/4 | 3/4 | 0/5 | 4/4 | 4/4 | 0/5 |
| $1.5 \times 10^4$ | 2/4 | 1/4 | 0/5 | 2/4 | 2/4 | 0/5 | 4/4 | 4/4 | 0/5 |
| $1.5 \times 10^3$ | 0/4 | 0/4 | 0/5 | 0/4 | 0/4 | 0/5 | 0/4 | 0/4 | 0/5 |
| SC frequency | 1 in 117,391 | 1 in 161,651 | 1 in 1,178,112 | 1 in 70,160 | 1 in 70,160 | 1 in 1,178,112 | 1 in 6,256 | 1 in 6,256 | 1 in 1,178,112 |

**Figure 7.** Inhibition of Notch and GLI1 enhances the effect of carboplatin in African American (AA)-triple negative breast cancer (TNBC); Notch1 and GLI1 are overexpressed in AA-TNBC tissues in comparison to TNBC tissues from White American (WA) women. (**A**) Schematic outline of the in vivo limiting dilution assay. (**B**) Tumor volume of secondary tumors established with 1.5 × 10⁶ tumor-dissociated cells from tumors formed in mammary fat pads of SCID/NOD mice implanted with HCC1806 cells and treated with vehicle, carboplatin (Carbo), or carboplatin + GANT-16 + DAPT (CGD).

*Figure 7 continued on next page*

*Figure 7 continued*

Secondary xenografts remain untreated. (**C**) Plots show Kaplan–Meier curves (blue – vehicle, orange – carboplatin, gray – CGD) for time to detect tumors in mice bearing secondary tumors. (**D**) Tumor incidence at weeks 2, 3, and 4 of secondary transplants of tumor-dissociated cells from tumors formed with HCC1806 cells and treated with vehicle, carboplatin (Carbo), or carboplatin + GANT-16 + DAPT (CGD) at limiting dilutions. The tumors/ numbers of mice/group are shown. The bottom row indicates the estimated breast tumor-initiating/stem cell (SC) frequencies. (**E**) Aldeflour assay using tumor-dissociated cells from secondary tumors formed in in vivo limiting dilution. *p ≤ 0.05. (**F**) Flow cytometry analysis of tumor-dissociated cells from secondary tumors formed in in vivo limiting dilution using ALDH1 staining. **p ≤ 0.01. (**G**) Expression of GLI1 and Notch intracellular domain (NICD) in TNBC tissue samples from AA and WA women. Representative images of IHC analysis of NICD and GLI1 expression in TNBC tissue samples from AA and WA women. Images are scored for staining intensity of cells as 0, 1, 2, 3 representing no, mild, moderate, or high staining intensity. (**H**) Graphical representation of GLI1 expression in TNBC tissues from AA women (n = 25) and WA women (n = 46) p < 0.001. (**I**) Graphical representation of NICD expression in TNBC tissues from AA women (n = 25) and WA women (n = 46) p < 0.001.

The online version of this article includes the following figure supplement(s) for figure 7:

**Figure supplement 1.** Reduced expression of Notch intracellular domain (NICD) and GLI1 in African American (AA)-triple negative breast cancer (TNBC) secondary tumors treated with a combination of carboplatin, GANT61, and DAPT.

**Figure supplement 2.** Inhibition of Notch1 and GLI1 along with carboplatin treatment inhibits stemness African American (AA)-triple negative breast cancer (TNBC) secondary tumors.

## GLI1 and NICD are highly expressed in TNBC tissues from AA women compared to WA women

We further corroborated the results from our in silico, in vitro, and in vivo studies using clinical samples. This study involved a total of 25 AA and 46 WA-TNBC samples (*Table 1*). Immunohistochemical staining of AA- and WA-TNBC tissues showed nuclear localization of NICD and GLI1 in tumor tissues with varying degrees of heterogeneity (*Figure 7G*). Nuclear intensity of NICD and GLI1 was scored using modified histochemical-score (H-score) system (described in Materials and methods). Significantly higher GLI1 (p < 0.001) and NICD (p < 0.001) expression was observed in AA-TNBC tissues compared to WA-TNBC tissues (*Figure 7H, I*, *Table 2*). These results explicitly show that TNBC tumors from AA women harbor elevated expression of oncogenic transcription factors, GLI1 and NICD, in comparison to WA women. Together, our study uncovers key molecular alterations that can potentiate a highly aggressive phenotype in AA-TNBC, and shows a novel therapeutic strategy whereby directly inhibiting these nodes can prove beneficial to target TNBC in AA women.

## Discussion

Existence of racial disparity in breast cancer-related mortality among AA and WA women cannot be completely explained by higher prevalence of TNBC, the most aggressive subtype of breast cancer, in AA women along with multiple socioeconomic features. We noted elevated incidence of TNBC and lower survival over all age-groups along with increased frequency of higher-grade tumors in AA women in SEER database and then embarked on investigating whether the TNBC arising in AA women are intrinsically wired to support rapid growth and metastasis. A direct comparison of AA-TNBC and WA-TNBC cells reveals several distinct features of AA-TNBC cells that advance our understanding in this regard. Particularly significant is that AA-TNBC cells inherently harbor a highly migratory and stemness-rich phenotype which is supported by the enrichment of epithelial-mesenchymal transition (EMT) and metastasis pathway genes. Indeed, cancer stem-like cells obtained from AA-TNBC cell line show increased self-renewal (*Yin et al., 2017*). Also, 48% TNBC in African women exhibit high ALDH+ population compared to non-African women where only 19–30% TNBC show high ALDH + population (*Ginestier et al., 2007*; *Nalwoga et al., 2010*). Higher stemness in AA-TNBC tumors may also explain significantly higher recurrence and worse overall survival in AA women with TNBC (*Doepker et al., 2018*; *Siddharth and Sharma, 2018*). Mechanistic evaluations identify elevated levels of Notch1 and GLI1 pathway genes in AA-TNBC cells and a positive correlation is observed between GLI1 gene signature and Notch1 gene signature in AA-TNBC tumors in contrast to WA-TNBC tumors. Few recent studies have investigated various biological aspects of racial disparity in TNBC. A comprehensive analysis of AA- and WA-TNBC tumors report that mutational landscape cannot explain the racial disparity between AA- and WA-TNBC as they have similar mutational profile (*Ademuyiwa et al., 2017*). AA women exhibit higher tumor-infiltrating CD8[+] T cell density compared to white women indicating

**Table 1.** Summary of patient characteristics: overall and by race.

| Variables | Black | White | p-Value | Total |
|---|---|---|---|---|
| Sample size | N = 25 | N = 46 | | N = 71 |
| Age – median (range) | 51 (33, 71) | 54 (34, 84) | 0.059 | 53.5 (33, 84) |
| Stage at diagnosis – n (%) | | | | |
| Distant | 0 (0) | 3 (7.7) | 0.005 | 3 (5.4) |
| Localized | 6 (35.3) | 27 (69.2) | | 33 (58.9) |
| Reg, DirEx | 2 (11.8) | 0 (0) | | 2 (3.6) |
| Reg, Nodes | 9 (52.9) | 8 (20.5) | | 17 (30.4) |
| Unstageable | 0 (0) | 1 (2.6) | | 1 (1.8) |
| Unknown | 8 | 7 | | 15 |
| Pathological T stage – n (%) | | | | |
| T1 | 0 (0) | 6 (27.3) | 0.075 | 6 (19.4) |
| T1c | 1 (11.1) | 4 (18.2) | | 5 (16.1) |
| T2 | 6 (66.7) | 12 (54.5) | | 18 (58.1) |
| T3 | 1 (11.1) | 0 (0) | | 1 (3.2) |
| T4d | 1 (11.1) | 0 (0) | | 1 (3.2) |
| Unknown | 16 | 24 | | 40 |
| Pathological N stage – n (%) | | | | |
| N0 | 5 (55.6) | 14 (66.7) | 0.478 | 19 (63.3) |
| N1 | 3 (33.3) | 4 (19) | | 7 (23.3) |
| N2 | 0 (0) | 1 (4.8) | | 1 (3.3) |
| N3 | 1 (11.1) | 0 (0) | | 1 (3.3) |
| NX | 0 (0) | 2 (9.5) | | 2 (6.7) |
| Unknown | 16 | 25 | | 41 |
| Pathological M stage – n (%) | | | | |
| M0 | 5 (71.4) | 9 (60) | >0.99 | 14 (63.6) |
| M1 | 0 (0) | 2 (13.3) | | 2 (9.1) |
| MX | 2 (28.6) | 4 (26.7) | | 6 (27.3) |
| Unknown | 18 | 31 | | 49 |
| Histological grade – n (%) | | | | |
| ModDiff-LowG | 4 (23.5) | 12 (33.3) | 0.676 | 16 (30.2) |
| PoorDif-MedG | 13 (76.5) | 23 (63.9) | | 36 (67.9) |
| WellDif | 0 (0) | 1 (2.8) | | 1 (1.9) |
| Unknown | 8 | 10 | | 18 |
| Stage – n (%) | | | | |
| I | 1 (8.3) | 9 (42.9) | 0.077 | 10 (30.3) |
| IIA | 3 (25) | 6 (28.6) | | 9 (27.3) |
| IIB | 4 (33.3) | 2 (9.5) | | 6 (18.2) |
| IIIA | 1 (8.3) | 0 (0) | | 1 (3) |
| IIIB | 1 (8.3) | 0 (0) | | 1 (3) |

*Table 1 continued on next page*

*Table 1 continued*

| Variables | Black | White | p-Value | Total |
|---|---|---|---|---|
| IV | 2 (16.7) | 4 (19) | | 6 (18.2) |
| Unknown | 13 | 25 | | 38 |
| Tumor size – mean (SD) | 3.28 (1.39) | 2.07 (0.94) | 0.011 | 2.46 (1.23) |
| Nodal status – n (%) | | | | |
| Negative | 6 (27.3) | 26 (66.7) | 0.004 | 32 (52.5) |
| Positive | 16 (72.7) | 13 (33.3) | | 29 (47.5) |
| Unknown | 3 | 7 | | 10 |
| LN, primary, benign – n (%) | | | | |
| LN | 4 (16.7) | 4 (9.5) | 0.448 | 8 (12.1) |
| Primary | 20 (83.3) | 38 (90.5) | | 58 (87.9) |
| Unknown | 1 | 4 | | 5 |

a potential involvement of immune landscape in racial disparity. Although no clear association is observed between CD8$^+$ T cell density and overall survival, higher CD8$^+$ T cell density in AA-TNBC associate with better overall survival which is paradoxical (*Abdou et al., 2020*). Interestingly, exosome analysis shows higher expression of exosomal-Annexin 2 (exo-AnxA2) in the sera of AA women with TNBC indicating its potential role in angiogenesis and tumor promotion (*Chaudhary et al., 2020*). On the other hand, proteome profiling of AA- and WA-TNBC exhibit very few alterations in proteins with HSP71 and HNRNP A2/B1 among the most altered proteins (*Torres-Luquis et al., 2019*). AA- and WA-TNBC tumors show differential expression of Kaiso, which has also been linked with increased TNBC aggressiveness (*Bassey-Archibong et al., 2017*). Although some of these studies indicate molecular differences between AA- and WA-TNBC, identification of key nodes that can trigger a molecular cascade to alter tumor progression has not been investigated.

Our study uncovers that AA-TNBC cells harbor intrinsically elevated levels of GLI1 and Notch1 in comparison to WA-TNBC cells, a finding validated in TNBC tissues from AA and WA women. Of importance, GLI1 and Notch1 are transcription factors capable of mediating the transactivation of numerous responsive genes and modulating the biological outcome in AA-TNBC. GLI1 is overexpressed in BCSCs characterized by CD44$^+$/CD24$^-$ phenotype compared to other cancer cells (*Liu et al., 2006*) and is important for EMT and metastatic progression (*Colavito et al., 2014*; *Riaz et al., 2018*). Inhibiting sonic hedgehog pathway and GLI1 nuclear localization abrogates CD44$^+$/CD24$^-$ positive cell population in TNBC (*Koike et al., 2017*; *Yang et al., 2016*). GLI1 also associates with increased angiogenesis in TNBC as GLI1 overexpression pairs with VEGFR2 and its inhibition reduces tumor angiogenesis (*Di Mauro et al., 2017*). Although expressed in many subtypes of breast cancer, GLI1 preferentially promotes growth of ER-negative breast cancer cells and is associated with poorer survival with TNBC compared to ER-positive disease (*Xu et al., 2010*). Previous studies have reported a breast cancer subtype-specific relationship between GLI1 and various aspects of stemness and metastatic progression. Our results present that GLI1 is inherently expressed at a higher level in AA-TNBC which also harbor a highly migratory and stemness-rich phenotype in comparison to WA-TNBC. Although Notch1 can act as an oncogene as well as a tumor suppressor depending on the cellular context (*Lobry et al., 2014*), its overexpression clearly associates with poor survival in TNBC. Higher growth, EMT, and stemness underlie aggressive metastatic progression of TNBC and worse overall survival. Notch1 can regulate these properties

**Table 2.** H-scores by race.

| Variables | Black | White | p-Value | Total |
|---|---|---|---|---|
| Log(GLI1 H-score) – mean (SD) | 5.41 (0.09) | 4.98 (0.17) | <0.001 | 5.13 (0.25) |
| Log(NICD H-score) – mean (SD) | 5.4 (0.1) | 5.02 (0.12) | <0.001 | 5.15 (0.22) |

as it plays an important role in maintenance of BCSCs, EMT, as well as metastasis (*Kim et al., 2019*; *Xie et al., 2017*). Overexpression of Notch1 overlaps with ALDH1 expression in breast cancer tissues and significantly correlates with high grade, metastasis, and TNBC (*Zhong et al., 2016*). A recent study shows a positive correlation between EMT-related genes and Notch1 activation in TNBC tumors (*Miao et al., 2020*). Positive association between Notch1 and EMT activator-melanoma cell adhesion molecule also denotes its role in EMT progression in TNBC, and accordingly, Notch1 inhibition reverses EMT (*Zeng et al., 2020*). Our data uncovers an additional layer of complexity pertaining to Notch1 and racial disparity in TNBC, and presents a greater functional impact of Notch1 in AA-TNBC compared to WA-TNBC.

Another interesting observation of our study is the positive correlation between GLI1 gene signature and Notch1 gene signature in AA-TNBC tumors in contrast to WA-TNBC tumors. Although a previous study reported the upregulation of Notch-responsive gene HES1 via sonic hedgehog in multipotent mesodermal cells (*Ingram et al., 2008*) and co-targeting Notch and hedgehog depletes docetaxel-resistant cells in hormone refractory prostate cancer (*Domingo-Domenech et al., 2012*), functional interactions between these important developmental pathways have been largely unexplored in breast cancer. We present a functional convergence between hedgehog/GLI and Notch pathways as their effector transcription factors, GLI1 and Notch1, exhibit nuclear co-localization and physical interaction via co-immunoprecipitation in AA-TNBC. Indeed, combined inhibition of GLI1 and Notch1 using GANT61 and DAPT exhibits synergistic inhibition of AA-TNBC cells indicating that targeting GLI1 and Notch1 in combination with chemotherapy may prove useful. TNBC are primarily treated with chemotherapy regimens but are marred with poor response and recurrence (*Marra et al., 2020*). In this context, regimens combining chemotherapy with GANT61 and DAPT not only show significant AA-TNBC tumor inhibition but also abrogate stem cell population. Several preclinical studies have examined the efficacy of notch pathway inhibition in cancer using anti-Notch monoclonal antibodies, peptide inhibitors, and γ-secretase inhibitors (*Groth and Fortini, 2012*; *Massard et al., 2018*; *Ran et al., 2017*). Various inhibitors targeting hedgehog/GLI pathway including SMO inhibitors like vismodegib and sonidegib as well as GLI1 inhibitors like GANT58 and GANT61 are under preclinical investigation (*Bhateja et al., 2019*). A phase 1, interventional, non-randomized clinical trial examined the pharmacological and pharmacodynamics of Notch inhibitor MK-0752 and reported the preferred scheduling of MK-0752 which modulated Notch gene signature. The study reported the clinical benefit as some patients on MK-0752 exhibited stable disease for longer duration (*Krop et al., 2012*). Combining escalating doses of MK-0752 with docetaxel showed benefit for patients with locally advanced or metastatic breast cancer (*Schott et al., 2013*). However, combining Notch inhibitor LY3039478 with additional anticancer agents (LY3023414, taladegib, or abemaciclib) did not yield superior results for patients with advanced or metastatic solid tumors (*Azaro et al., 2021*). Few clinical trials are currently investigating the efficacy of AL101 (an inhibitor of gamma secretase-mediated Notch signaling) (*ClinicalTrials.gov Identifier: NCT04461600*), and CB-103 (inhibitor of CSL-NICD gene transcription factor complex) (*ClinicalTrials.gov Identifier: NCT03422679*). Clinical advances with Notch1 and GLI1 inhibitors will guide the suitable dosing schedules for targeting AA-TNBC. Also, further studies using AA-TNBC PDXs and clinically viable Notch and GLI1 inhibitors will help optimize dose and treatment schedules and strengthen the clinical utility of combination regimens utilizing GLI1/Notch1 inhibitors along with standard chemotherapy. Our results provide preclinical support for the clinical investigations where TNBC in AA women can be effectively treated via co-targeting the actionable key molecular nodes.

## Conclusions

The results presented here explicitly demonstrate that AA-TNBC cells are highly migratory with elevated self-renewal potential and stem-like phenotype compared to WA-TNBC. We uncover the aberrant activation and functional interaction of Notch1 and GLI1 in AA-TNBC. In addition, combining chemotherapy which inhibits bulk tumor with Notch1 and GLI1 inhibitors that abrogate stemness display an excellent synergy in inhibiting AA-TNBC, hence providing a potent therapeutic option for abrogating TNBC in AA women. As we move toward personalized therapy for cancer management, race of the individual has presented another variable to consider, especially to counter racial disparity.

Understanding the unique biology of TNBC arising in AA women and guiding the therapeutic regimens on its basis is very challenging, yet very exciting goal.

## Materials and methods

### Key resources table

| Reagent type (species) or resource | Designation | Source or reference | Identifiers | Additional information |
|---|---|---|---|---|
| Antibody | Nanog (Rabbit monoclonal) | Cell Signaling Technology | 4903S | 1:1000 dilution |
| Antibody | Oct4 (Rabbit polyclonal) | Cell Signaling Technology | 2750S | 1:1000 dilution |
| Antibody | Sox2 (Mouse monoclonal) | Santa Cruz Biotechnology Inc | sc-365964 | 1:1000 dilution |
| Antibody | KLF4 (Rabbit monoclonal) | Cell Signaling Technology | 12173S | 1:1000 dilution |
| Antibody | cMyc (Rabbit monoclonal) | Cell Signaling Technology | 5605S | 1:1000 dilution |
| Antibody | NICD (IHC) (Rabbit monoclonal) | Cell Signaling Technology | 4147S | 1:1000 dilution |
| Antibody | Notch 1 (Mouse monoclonal) | Santa Cruz Biotechnology Inc | sc-376403 | 1:1000 dilution |
| Antibody | JAGGED (Mouse monoclonal) | Santa Cruz Biotechnology Inc | sc-390177 | 1:1000 dilution |
| Antibody | Ki67 (Rabbit monoclonal) | Cell Signaling Technology | 9027S | 1:1000 dilution |
| Antibody | GLI1 (Mouse monoclonal) | Santa Cruz Biotechnology Inc | sc-515781 | 1:1000 dilution |
| Antibody | SHH (Mouse monoclonal) | Santa Cruz Biotechnology Inc | sc-365112 | 1:1000 dilution |
| Antibody | FOXM1 (Mouse monoclonal) | Santa Cruz Biotechnology Inc | sc-376471 | 1:1000 dilution |
| Antibody | HEY1 (Rabbit polyclonal) | ABclonal Technology, Woburn, MA | A16110 | 1:1000 dilution |
| Antibody | HES1 (Rabbit monoclonal) | Cell Signaling Technology | 11988S | 1:1000 dilution |
| Antibody | Actin (Mouse monoclonal) | Sigma-Aldrich, St Louis, MO | A5441 | 1:10,000 dilution |
| Antibody | Anti-ALDH (FACS) (Mouse monoclonal) | BD Biosciences, San Jose, CA | 611194 | 1:500 dilution |
| Antibody | CD49f-APC (FACS) (Rat monoclonal) | BioLegend, San Diego, CA | 313616 | 1:500 dilution |
| Antibody | CD24-FITC (FACS) (Mouse monoclonal) | BD Biosciences, San Jose, CA | 555427 | 1:500 dilution |
| Antibody | CD44-PE (FACS) (Mouse monoclonal) | BD Biosciences, San Jose, CA | 550989 | 1:500 dilution |
| Chemical compound, drug | DAPT (for in vitro assays) | Sigma-Aldrich, St Louis, MO | D5942 | |
| Chemical compound, drug | DAPT (for in vivo assays) | Selleck Chemicals, Houston, TX | S2215 | |
| Chemical compound, drug | GANT61 | Selleck Chemicals, Houston, TX | S8075 | |

### Cell culture and reagents

Human TNBC cell lines MDAMB468, HCC1806, HCC1569, Hs578t, BT549, MDAMB231, and HCC1937 (*Wu et al., 2015*) were procured from American Type Culture Collection (ATCC, Manassas, VA), revived from early passage liquid nitrogen vapor stocks as required and maintained at 37°C in 5% $CO_2$ and 95% humidity. All cells were authenticated via short tandem repeat testing. No mycoplasma contamination was noted. DAPT and GANT61 were procured from Sigma-Aldrich, St. Louis, MO, and

Selleck Chemicals, Houston, TX. For Western blot, immunoprecipitation, immunofluorescence and immunohistochemistry, anti-ALDH1A, anti-Nanog, anti-Oct4, anti-KLF4, anti-c-MYC, anti-NICD, anti-JAGGED, and anti-Ki67 antibodies were purchased from Cell Signaling Technology, Beverly, MA. Antibodies anti-GLI1, anti-Sox2, anti-SHH, anti-FOXM1, and anti-HES1 were procured from Santa Cruz Biotechnology Inc Anti-HEY1 was purchased from ABclonal Technology, Woburn, MA. Mouse monoclonal β-Actin was procured from Sigma-Aldrich, St. Louis, MO. Horseradish peroxidase conjugated goat anti-rabbit IgG, goat anti-mouse IgG, and donkey anti-goat IgG were purchased from Sigma-Aldrich, St. Louis, MO. 3-(4,5-Dimethylthiazol-2-yl)–2,5-diphenyltetrazolium bromide (MTT) were procured from Sigma-Aldrich, St Louis, MO. Chemiluminescent peroxidase substrate was procured from GE Healthcare, UK.

## In silico data analysis

Normalized mRNA data from TCGA-BRCA cohort was retrieved from the Broad GDAC Firehose (https://gdac.broadinstitute.org/) and processed using Microsoft excel. We searched publicly available Gene Expression Omnibus datasets with 'TNBC' as query term. We downloaded the data of GSE46581 (*Lindner et al., 2013*) and performed GSEA for enriched gene set of Hallmark EMT and metastasis pathways in cancer in the AA-TNBC samples compared to WA-TNBC samples using GSEA software. *SEER data analysis*: Data were analyzed using US Surveillance, Epidemiology, and End Results (SEER database) (https://seer.cancer.gov/statfacts/html/breast.html). Graphical data of recent trends of incidence rates from 2010 to 2016 and percentage of surviving population from TNBC with respect to age, race, and ethnicity, respectively, was generated using SEER online tool. *Survival analysis*: Association between GLI1, GLI1-target genes, Notch1 and Notch1-target genes, and recurrence-free survival for patients with TNBC was examined using survival data using Cox proportional hazards regression and by plotting Kaplan–Meier survival plots (*Nagy et al., 2018*). *Correlation analysis*: Data collection: TCGA normalized reads were downloaded from the FireHose-Broad Institute database. Patients were classified based on the negative status of ER, PR, and HER2 and divided into two groups: AA and WA. ssGSEA: Individual reads of each patient were used in ssGSEA. Analysis was performed through the GenePattern Platform using GLI1 and Notch1 signature. This signature was specific and submitted to user gene set option. Further, ssGSEA enrichment score of a pathway for all patients and gene-specific normalized expression count of each patient were used to do Pearson correlation analysis on GraphPad Prism 8.0 and the data is represented as scatter plots.

## Cell viability and clonogenicity

### MTT

The anchorage-dependent viability of AA-TNBC and WA-TNBC cells was measured by estimating the reduction of 3-(4,5-dimethylthiazol-2-yl)–2,5-(diphenyltetrazolium bromide) in a MTT assay (Sigma) following manufacturer's protocol. The percentages of live cells were calculated and plotted using Prism software (GraphPad Prism, CA, and Excel). Cells were treated with DAPT, GANT61, doxorubicin, and/or carboplatin as indicated in the figures. Using Microsoft excel and Compusyn (Compusyn Inc, Paramus, NJ) software, the IC50 and combination index was calculated, respectively.

### Clonogenicity

TNBC cells were subjected to clonogenicity following standard protocol (*Muniraj et al., 2019*), treated as indicated in the figures and the colonies were stained with crystal violet (0.1% in 20% methanol). Colonies containing >50 normal-appearing cells were counted and pictures were taken using a digital camera.

### Trypan blue exclusion assay

TNBC cells were harvested using trypsin (0.2%), stained with trypan blue (Sigma-Aldrich, St Louis, MO) and counted using a hemocytometer under the phase contrast microscopy. All experiments were performed thrice in triplicates.

## Scratch-migration, spheroid-migration, and transwell-migration assay

### Scratch-migration assay

AA-TNBC and WA-TNBC cells were trypsinized and 70 µl of the cell suspension ($3 \times 10^5$ cells/ml) was seeded in each well of the ibidi Culture-Insert 2 well in 35 mm cell culture dishes overnight and allowed to form monolayer. The culture-inserts were removed using sterile tweezers; the cell monolayer was washed with PBS to remove the cell debris and unattached cells and fresh medium was added. Plates were photographed immediately and migration of cells was followed for various time intervals. Wound closure was quantified from distance between edges using Leica ImageScope software and plotted using GraphPad Prism 5 software.

### Spheroid-migration assay

Migration of cells from tumor cell spheroid was examined using our previously published protocol (*Avtanski et al., 2015*). TNBC cells ($1.5 \times 10^4$) were resuspended in 0.5% agar-coated 96-well plates and cultured on an orbital shaker for 48 hr at 37°C in a humidified atmosphere of 5% $CO_2$ for the formation of tumor spheroids. Intact tumor spheroids were selected and transferred onto 12-well plates followed by incubation for 48 hr to allow migration of tumor cells from the spheroids. Fixed spheroids were stained with crystal violet, and the migrated cells were observed microscopically, quantified using Leica ImageScope software, and plotted using GraphPad Prism 5 software.

### Transwell-migration assay

Briefly, $2 \times 10^5$ cells were seeded in serum-free media in the upper chamber of transwell inserts in 12-well plates. The lower chambers were filled with serum supplemented media and incubated for 48 hr. Migrated cells were fixed, stained with 0.05% crystal violet, imaged using microscope, quantified using Leica ImageScope software, and plotted using GraphPad Prism 5 software.

### Matrigel invasion assay

Invasion potential of TNBC cells was examined using Matrigel invasion assay according to our previously published protocol (*Avtanski et al., 2014*). Tumor-dissociated cells ($2 \times 10^4$) were seeded in the Matrigel invasion chamber from BD Biocoat Cellware (San Jose, CA). Cells invaded through Matrigel were fixed, stained with crystal violet (0.1% in 20% methanol), imaged using microscope, quantified using Leica ImageScope software, and plotted using GraphPad Prism 5 software.

### Mammosphere assay

Mammosphere formation potential of AA-TNBC and WA-TNBC was examined using liquid and solid mammosphere assay (*Avtanski et al., 2016*). For *liquid mammosphere* assay, $5 \times 10^3$ AA-TNBC or WA-TNBC cells were seeded in 2 ml of liquid mammosphere media supplemented with 1% penicillin/streptomycin, B27 (1:50, Invitrogen-Life Technologies), 5 µg/ml insulin, 1 µg/ml hydrocortisone (Sigma), 20 ng/ml EGF (R&D Systems), 20 ng/ml basic fibroblast growth factor in 30 mm ultra-low attachment plates. Cells were allowed to grow for 7 days. For *solid mammosphere* assay, $5 \times 10^3$ AA-TNBC or WA-TNBC cells were suspended in 2 ml mammosphere medium containing methylcellulose, plated on ultra-low attachment plates, and incubated for 7 days. Cultures were observed under microscope and spheres (>50 µm) were counted manually in random fields.

## Transfection, protein isolation, sub-cellular fractionation, co-immunoprecipitation, and Western blotting

Cells were transfected with hGLI1-Flag3x (Addgene, Cat#84922) using Lipofectamine 2000 (Thermo Fisher Scientific) following manufacturer's instructions. Whole cell lysates from AA-TNBC and WA-TNBC cells were prepared using modified RIPA buffer. Cytoplasmic and nuclear extracts were prepared using NE-PER Nuclear and Cytoplasmic Extraction Reagent kit (Thermo Fisher Scientific, Waltham, MA). Co-immunoprecipitation of GLI1 and NICD from wild-type and hgli1flag overexpressing HCC1569 cells was performed following our published protocol (*Saxena et al., 2008*) using anti-GLI1 and anti-NICD antibodies for immunoprecipitation followed by immunoblotting with anti-NICD antibody. Equal amount of protein was resolved on sodium-dodecyl sulfate polyacrylamide

gel, transferred onto PVDF membrane and immunoblotted using specific antibodies. Western blot quantification was performed using ImageJ.

## Luciferase assay

For luciferase reporter assay, cells were seeded in 12-well plate and were transfected with 4xCSL-luciferase plasmid (a gift from Raphael Kopan [Addgene plasmid # 41726; http://n2t.net/addgene: 41726]) using Lipofectamine 2000 following manufacturer's instructions. To evaluate the effect of Gli1 overexpression on Notch1 transactivation, 4xCSL-luciferase plasmid was co-transfected with hGLI1-Flag3x gene. After 48 hr of transfection, cells were harvested and lysed; the renilla and luciferase activity was measured using dual-luciferase reporter assay system (Promega) following manufacturer's instruction.

## Immunocytochemical staining, immunohistochemistry, and imaging

TNBC cells ($2 \times 10^5$ cells/well) were resuspended in eight-well chamber slides (Nunc, Rochester, NY). Untreated and treated TNBC cells were subjected to immunocytochemical staining following the published protocol (*Nagalingam et al., 2012*). Fixed cells were permeabilized using 0.1% Triton-X-100 followed by 16 hr incubation with primary antibody at 1:100 dilution in 3% BSA. Cells were then incubated with FITC/TRITC-tagged secondary antibody and examined using a Leica E800 fluorescent microscope. Images were captured at 63× magnification using oil immersion objective with Leica Elements software.

### Immunohistochemistry

Tumors were fixed in 10% formalin, paraffin-embedded, and sectioned. Tissues were probed with primary antibodies, anti- GLI1, anti-NICD, anti–Ki-67, anti-oct4, and anti-SOX2 antibodies, followed by HRP conjugated secondary antibody and developed using DAB peroxidase substrate kit (SK-4100, Vector Laboratories, CA). Images were captured with a Leica microscope at 20× magnification. Images were analyzed by Aperio ImageScope Software, Leica. Quantification of micrographs and IHC was done using Leica Aperio ImageScope, Leica Biosystems.

## Flow cytometry to detect cancer stem cell markers

The expression profile of CD44 (*anti-human#550989, BD Bioscience*), CD24 (*anti-human#555427, BD Biosciences*), and CD49f (*anti-human/mouse#313616, BioLegend*) in cultured AA-TNBC and WA-TNBC cells and tumor-dissociated cells were analyzed by flow cytometry. In brief, $1 \times 10^6$ cells were stained with respective antibodies following antibody-manufacturer's specific protocol. Labeled cells were analyzed by BD FACS LSR II using FACS Diva 6.0 software.

### ALDEFLOUR assay

Tumor-dissociated cells were suspended in ALDEFLOUR assay buffer, incubated with activated ALDE-FLOUR reagent for 30 min at 37°C in dark and ALDH enzymatic activity was measured using the ALDE-FLOUR kit (Stem Cell Technologies, Vancouver, BC, Canada) following the manufacturer's protocol. Diethylaminobenzaldehyde (ALDH inhibitor) was included as negative control.

## Orthotopic-xenograft model and limiting-dilution orthotopic-xenograft model

All animal experiments were according to Johns Hopkins ACUC. SCID-NOD mice (female, 6–8 weeks of age) were procured from SKCCC animal facility and maintained in-house. Exponentially growing HCC1806, HCC1937, or MDA-MB-231 cells ($5 \times 10^7$ cells in 100 µl Matrigel) were implanted in the fourth mammary fat pad on either side of the mouse. Tumor-bearing mice (n = 15/cell line) were divided into three treatment group (n = 5/group). Mice were treated with vehicle, DAPT (20 mg/kg of body weight, 5 days/week) + GANT61 (50 mg/kg of body weight, twice a week) combination, and doxorubicin (2 mg/kg of body weight, once a week) + carboplatin (50 mg/kg of body weight, single dose) combination. Tumor growth was regularly measured. At the end of experiment, tumors were excised and processed for further analysis. In another experiment, exponentially growing HCC1806 cells ($5 \times 10^7$ cells in 100 µl Matrigel) were implanted in the fourth mammary fat pad on either side of the mouse. Tumor-bearing mice (n = 35) were divided into seven treatment group (n = 5/group) and

treated intraperitoneally with vehicle, carboplatin (50 mg/kg of body weight, single dose), doxorubicin (2 mg/kg of body weight, once a week), DAPT (20 mg/kg of body weight, daily M-F), GANT61 (50 mg/kg of body weight, every other day), carboplatin + GANT61 + DAPT, and doxorubicin + GANT61 + DAPT throughout the experimental duration. Tumor growth was regularly measured. At the end of experiment, tumors were excised and processed for further analysis. For *limiting-dilution orthotopic-xenograft model* (*Sengupta et al., 2017*), tumor cells dissociated from primary tumors formed in vehicle, carboplatin, and carboplatin + GANT61 + DAPT treatment groups were injected at limiting dilutions ($1.5 \times 10^6$–$1.5 \times 10^3$), into the fourth and fifth mammary fat pads of immunocompromised SCID-NOD mice (female, 6–8 weeks of age) on both sides. Tumor incidence was regularly monitored. Stem cell frequency was calculated based on tumor incidence in all treatment groups using L-Calc software for LDA.

## TNBC tissue samples

Tissue microarray consisting of 25 AA- and 46 WA-TNBC specimens was generated at Yale Developmental Histology Lab using TNBC specimens that were diagnosed at the Yale-New Haven Hospital, New Haven, CT, between 1996 and 2004. TNBC status (absence of ER, PR, and HER2) was determined by IHC at the Yale Developmental Histology Lab. *Table 1* shows a summary of the clinicopathological features of TNBC specimens included in the TMA. Tissue sections (5 µm) were subjected to immunohistochemical analysis using anti-NICD and anti-GLI1 antibodies and scored in a blinded manner as 0, 1, 2, or 3 representing no, mild, moderate, or high staining intensity. Total score was generated using modified histochemical (H) score system spanning values from 0 to 300: 3× (percentage of cells with high intensity staining (3+)) + 2× (percentage of cells with moderate intensity staining (2+)) + 1× (percentage of cells with mild intensity staining (1+)) for each specimen.

## Statistical analysis

Statistical analyses were exploratory. Analyses were done using R and GraphPad Prism 5. Data was transformed when necessary to achieve normality. One-way ANOVA was conducted for overall comparison across multiple groups. Pairwise comparisons were done using Student's t-test with Bonferroni correction. Kaplan–Meier survival curves were reported for survival outcomes and log-rank test was conducted to compare survival outcomes between groups. Results were considered to be significant if p-value < 0.05.

## Additional information

### Funding

| Funder | Grant reference number | Author |
|---|---|---|
| National Cancer Institute, NIH | R01CA204555 | Dipali Sharma |

The funders had no role in study design, data collection and interpretation, or the decision to submit the work for publication.

### Author contributions

Sumit Siddharth, Data curation, Formal analysis, Investigation, Methodology, Validation, Writing - original draft; Sheetal Parida, David Lim, Chenguang Wang, Balazs Gyorffy, Juliet M Daniel, Investigation, Methodology; Nethaji Muniraj, Arumugam Nagalingam, Formal analysis, Investigation, Methodology; Shawn Hercules, Investigation; Dipali Sharma, Conceptualization, Data curation, Formal analysis, Funding acquisition, Investigation, Methodology, Project administration, Resources, Software, Supervision, Validation, Visualization, Writing - review and editing

### Author ORCIDs

Dipali Sharma http://orcid.org/0000-0002-5032-974X

## Ethics

This study was performed in strict accordance with the recommendations in the Guide for the Care and Use of Laboratory Animals of the National Institutes of Health. All of the animals were handled according to approved institutional animal care and use committee of Johns Hopkins ACUC. The animal protocol was approved by Johns Hopkins ACUC. All animal studies were conducted according to the animal protocol M019M475 approved by JHU ACUC. No clinical data.

## Decision letter and Author response

Decision letter https://doi.org/10.7554/eLife.70729.sa1
Author response https://doi.org/10.7554/eLife.70729.sa2

## Additional files

### Supplementary files

• Transparent reporting form

### Data availability

All data generated or analysed during this study are included in the manuscript and supporting files.

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
