## [Editor Report]

This paper represents a fundamental contribution to our understanding of potential differences in the incidence of triple negative breast cancer in Caucasian and African American women. In addition, the findings provide potential insights relating to the relative effectiveness of therapy in these populations, and suggest novel therapeutic approaches.

---

## [Decision Letter]

**Decision letter after peer review:**

Thank you for submitting your article "Concomitant activation of GLI1 and Notch1 contributes to racial disparity of triple negative breast cancer progression" for consideration by *eLife*. Your article has been reviewed by 3 peer reviewers, one of whom is a member of our Board of Reviewing Editors, and the evaluation has been overseen by Maureen Murphy as the Senior Editor. The reviewers have opted to remain anonymous.

Essential revisions:

1. The major issue is that the treatments ( DAPT, GANT61, alone and in combination as well as in combination with doxorubicin or carboplatin) are evaluated solely in the cell lines from African American women. There are two problems here. One is related to all of the data that precedes the chemotherapy testing that suggests that the combinations may not (or should not?) be equally effective in breast tumors of Caucasian women. This may not prove to be the case, which would raise issues as to the relationship of the two sets of findings. The other is relevant in terms of whether some of the proposed therapies should ultimately also be investigated in clinical trials that include Caucasian women, should this work reach that point.

This is not to say that all of the chemotherapy testing should be repeated in the breast tumor cells from Caucasian women. But the authors are obligated to at least perform some of the key experiments in these cell lines.

2. The drug efficacy studies appear to be poorly rationalized. In Figure 6: The authors show that a triple combination of drug is superior than a single drug alone. This is not surprising, and I can think of few occasions where 3 agents were not superior to one. It is unclear why the authors are testing combinations with Dox or carbo. The real question is, does the Notch/GLI1 inhibitor combination fare better in AA BC tumors than the current standard of care, such as combined doxorubicin plus taxol? And, does it fare better in AA tumors compared to WA tumors? Such findings would greatly strengthen the impact of the work.

3. Figure 5, the interaction between Notch and GLI, is the weakest part of the manuscript. The authors should perform IP-western on endogenous proteins in both directions, and confirm using proximity ligation analyses. Also there needs to be a functional assay: does Notch enhance transactivation by GLI, and/or vice versa? Co-IF of two proteins does not indicate interaction; can the authors show whether these two proteins interact directly, for example using recombinant protein? Finally, the authors should provide insight as to why WA TNBC cells show very little NICD compared to AA TNBC cells even after IP.

4. The presented data in Figure 2G are not very impressive, better quality images may be presented. It is not clear whether Oct4, Nanog, and *SOX2* are localized in nucleus or cytoplasmic compartment.

5. The abstract does not cover all the key data presented in the manuscript. Similarly, the Discussion section also needs improvement by highlighting the key findings and their implications in the field. For example, first paragraph of the discussion is mostly repetition of the results.

6. How is ethnicity reported in TCGA? Did the authors use ancestry markers in order to validate ethnicity (WA vs AA)?

7. In Figure 7 it is not clear why the authors are adding carbo or dox; the real question is do Notch/GLI1 inhibitor combination cause reduced Aldefluor activity? Also the authors should shows some of the IHC data in the TMA.*Reviewer #1:*

The authors were attempting to determine the basis for the higher incidence of triple negative breast cancer in African American versus Caucasian women and reduced survival in the African American population. The studies presented are quite rigorous and largely convincing. The authors utilized both clinical data, cell culture approaches as well as experiments in tumor bearing animals to address these questions. They also identified potential new therapeutic approaches. The extensive experimental work and data generated largely support the conclusions presented.

This work is likely to have a significant impact in the field of breast cancer, in large part by providing a framework or roadmap as to how studies addressing these types of questions should be performed.

*Reviewer #2:*

The authors demonstrate that TNBC cells derived from AA patients are aggressive and show cancer stem-like phenotype compared to TNBC cells derived from White-American (WA). They observed enrichment as well as nuclear localization of GLI1 and Notch1 pathways in AA-TNBC cells compared to WA-TNBC cells. They further observed that GLI1 interacts with Notch1 in AA-TNBC cells and inhibition of GLI1 and Notch1 synergistically inhibits tumor growth, stemness, and cancer promoting functions. Clinically, they observed that GLI1 and cytoplasmic domain or Notch intracellular domain (NICD) are overexpressed in TNBC tissues from AA women. These observation are highly interesting and provide the molecular basis of breast cancer health disparity among American women. The presented data support the conclusion drawn in this manuscript, but some aspects of the manuscript need clarification and justification.

*Reviewer #3:*

This manuscript analyzes TCGA data from TNBC from African American (AA) and white American (WA) women. TCGA data indicates that there is a higher GLI and Notch signature in AA women, and analysis of a small subset of cell lines from AAM vs WA women (between 4-7 lines) suggests that GLI1 and Notch are overexpressed in breast cancer cell lines from AA as well. These findings are corroborated with tumor micro-array data, and this is a strength. Overall the conclusion that Notch and GLI are increased in AA TNBC is well supported by the data presented.

Weaknesses include the superficial nature of the description of the interaction between Notch and GLI1: it is unclear whether these proteins bind directly, and/or what the impact of this complex is on either Notch or GLI1 function. Additionally, the drug efficacy studies compare triple agents versus a single agent; it is not surprising that 3 agents work better than one.

---

## [Author Response]

Essential revisions:1. The major issue is that the treatments ( DAPT, GANT61, alone and in combination as well as in combination with doxorubicin or carboplatin) are evaluated solely in the cell lines from African American women. There are two problems here. One is related to all of the data that precedes the chemotherapy testing that suggests that the combinations may not (or should not?) be equally effective in breast tumors of Caucasian women. This may not prove to be the case, which would raise issues as to the relationship of the two sets of findings. The other is relevant in terms of whether some of the proposed therapies should ultimately also be investigated in clinical trials that include Caucasian women, should this work reach that point.This is not to say that all of the chemotherapy testing should be repeated in the breast tumor cells from Caucasian women. But the authors are obligated to at least perform some of the key experiments in these cell lines.

We appreciate the reviewers’ concerns. We have examined the combination treatment regimens in AA-TNBC as well as WA-TNBC cell lines. We observed that DAPT and GANT61 combination inhibited clonogenicity and cell viability of AA-TNBC (MDA-MB-468, HCC1569, HCC-1806) cells while WA-TNBC cells (BT549, MDA-MB-231 and HCC1937) did not show any inhibition of clonogenicity/cell viability in response to single/combination regimen of DAPT and GANT61. These new data are included in Figure 5G and Figure 5—figure supplement 2. In addition, we also evaluated the effectiveness of combination treatment with GLI1 and Notch1 inhibitors in comparison to Doxorubicin and Carboplatin combination in AA-TNBC and WA-TNBC tumors. AA-tumors showed better response to GLI1 and Notch1 inhibitors combination in comparison to chemo combination. In contrast, WA tumors showed no response to combination treatment with GLI1 and Notch1 inhibitors. These new data are included in (Figure 6—figure supplement 1, Figure 6A and Figure 6—figure supplement 2).

2. The drug efficacy studies appear to be poorly rationalized. In Figure 6: The authors show that a triple combination of drug is superior than a single drug alone. This is not surprising, and I can think of few occasions where 3 agents were not superior to one. It is unclear why the authors are testing combinations with Dox or carbo. The real question is, does the Notch/GLI1 inhibitor combination fare better in AA BC tumors than the current standard of care, such as combined doxorubicin plus taxol? And, does it fare better in AA tumors compared to WA tumors? Such findings would greatly strengthen the impact of the work.

We appreciate the reviewers’ concerns. We have evaluated the Notch/GLI1 inhibitor combination and chemotherapy combination (Doxorubicin and Carboplatin) in AA and WA tumors.

We investigated the effectiveness of combination treatment with GLI1 and Notch1 inhibitors in comparison to Doxorubicin and Carboplatin combination in AA-TNBC and WA-TNBC tumors. Mice harboring HCC1806, HCC1937 and MDA-MB-231-derived tumors were regularly treated with DAPT + GANT61 combination or Doxorubicin + Carboplatin combination and tumor progression was monitored. AA-TNBC (HCC1806) tumors showed more effective growth inhibition in response to DAPT + GANT61 in comparison to chemotherapy combination whereas WA-TNBC (HCC1937 and MDA-MB-231) tumors showed enhanced growth inhibition in response to Doxorubicin + Carboplatin combination. GANT61 + DAPT combination treatment did not reduce the growth of WA-TNBC cells-derived tumors (Figure 6—figure supplement 1). Tumor dissociated cells from WA-TNBC-HCC1937-derived tumors showed no/low reduction of the Aldehyde dehydrogenase activity in response to DAPT + GANT61 combination treatment (32.7%) in comparison to vehicle-treated group (36.5%) and DOX-CARBO group (31.9%). In contrast, AA-TNBC-HCC1806-derived tumors exhibited reduced Aldehyde dehydrogenase activity in DAPT + GANT61 treatment group (53.3%) compared to vehicle-treated group (77.5%) and DOX-CARBO (66.4%) (Figure 6A, Figure 6—figure supplement 2).

3. Figure 5, the interaction between Notch and GLI, is the weakest part of the manuscript. The authors should perform IP-western on endogenous proteins in both directions, and confirm using proximity ligation analyses. Also there needs to be a functional assay: does Notch enhance transactivation by GLI, and/or vice versa? Co-IF of two proteins does not indicate interaction; can the authors show whether these two proteins interact directly, for example using recombinant protein? Finally, the authors should provide insight as to why WA TNBC cells show very little NICD compared to AA TNBC cells even after IP.

We thank the reviewer for these suggestions. We have performed multiple experiments to strengthen the interaction between Notch1 and GLI1. WA TNBC cells show very little NICD compared to AA TNBC cells even after IP because WA TNBC cells have very low level of endogenous NICD in comparison to AA TNBC cells. We use same amount of input protein to immunoprecipitate NICD in both sets of cell lines.

We further evaluated the interaction between nuclear GLI1 and nuclear NICD endogenously present in AA-TNBC and WA-TNBC cells. In a coimmunoprecipitation assay, NICD was immunoprecipitated with GLI1 antibody in the nuclear extracts of AA-TNBC cells while low/no presence of NICD was observed in GLI1-immunoprecipitates in WA-TNBC cells (Figure 5C). In a reciprocal analysis, GLI1 was immunoprecipitated with NICD antibody in the nuclear extracts of AA-TNBC cells while low/no pulldown of GLI1 with Notch1 was observed in WA-TNBC cells (Figure 5C). Next, we investigated whether GLI1 overexpression influences transcriptional activity of NICD in AA-TNBC. Cells overexpressing GLI1 exhibited increased NICD transactivation in a luciferase assay (Figure 5D) and also, showed increased expression of Notch-responsive genes-Hes1 and Hey1 (Figure 5E). Notch1 overexpression also led to increased expression of GLI1-responsive gene-FOXM1 (Figure 5E). These new data are provided in Figure 5 C, D, and E.

4. The presented data in Figure 2G are not very impressive, better quality images may be presented. It is not clear whether Oct4, Nanog, and SOX2 are localized in nucleus or cytoplasmic compartment.

We appreciate the reviewers’ concerns. We have repeated these experiments and provided new images showing nuclear localization of cMyc, Oct4, Nanog and *Sox2* in AA TNBC (HCC1806, HCC1569 and MDA-MB-468) cells. We also included WA-TNBC (HCC1937, BT549 and Hs578t) cells. These new data have replaced images in Figure 2G.

5. The abstract does not cover all the key data presented in the manuscript. Similarly, the Discussion section also needs improvement by highlighting the key findings and their implications in the field. For example, first paragraph of the discussion is mostly repetition of the results.

We thank the reviewer for these suggestions. We have revised the abstract and Discussion section.

6. How is ethnicity reported in TCGA? Did the authors use ancestry markers in order to validate ethnicity (WA vs AA)?

TCGA data: TCGA data reports ethnicities as part of their clinicopathological parameters. We extracted these data from TCGA for our studies. TMA: The TMA was constructed using TNBC samples collected from 25 AA women and 46 WA women. The ethnicities were reported in their clinical charts. Cell lines: We acquired all the cell lines used in this study from ATCC. ATCC has clearly listed the ethnicities of the patients from whom these cell lines were derived. Authors did not use any ancestry markers.

7. In Figure 7 it is not clear why the authors are adding carbo or dox; the real question is do Notch/GLI1 inhibitor combination cause reduced Aldefluor activity? Also the authors should shows some of the IHC data in the TMA.

We thank the reviewer for these suggestions. We have examined Aldefluor activity in AA-TNBC and WA-TNBC tumors developed in mice treated with vehicle, Notch/GLI1 inhibitor combination and chemotherapy combination. Tumor dissociated cells from WA-TNBC-HCC1937-derived tumors showed no/low reduction of the Aldehyde dehydrogenase activity in response to DAPT + GANT61 combination treatment (32.7%) in comparison to vehicle-treated group (36.5%) and DOX-CARBO group (31.9%). In contrast, AA-TNBC-HCC1806-derived tumors exhibited reduced Aldehyde dehydrogenase activity in DAPT + GANT61 treatment group (53.3%) compared to vehicle-treated group (77.5%) and DOX-CARBO (66.4%) (Figure 6A, Figure 6—figure supplement 2). We have included the IHC data from TMA.

Reviewer #2 (Recommendations for the authors):The authors demonstrate that TNBC cells derived from AA patients are aggressive and show cancer stem-like phenotype compared to TNBC cells derived from White-American (WA). They observed enrichment as well as nuclear localization of GLI1 and Notch1 pathways in AA-TNBC cells compared to WA-TNBC cells. They further observed that GLI1 interacts with Notch1 in AA-TNBC cells and inhibition of GLI1 and Notch1 synergistically inhibits tumor growth, stemness, and cancer promoting functions. Clinically, they observed that GLI1 and cytoplasmic domain or Notch intracellular domain (NICD) are overexpressed in TNBC tissues from AA women. These observation are highly interesting and provide the molecular basis of breast cancer health disparity among American women. The presented data support the conclusion drawn in this manuscript, but some aspects of the manuscript need clarification and justification.

We thank the reviewer for the comments. We have provided more clarification and justification for all the aspects pointed by the reviewers.

Reviewer #3 (Recommendations for the authors):This manuscript analyzes TCGA data from TNBC from African American (AA) and white American (WA) women. TCGA data indicates that there is a higher GLI and Notch signature in AA women, and analysis of a small subset of cell lines from AAM vs WA women (between 4-7 lines) suggests that GLI1 and Notch are overexpressed in breast cancer cell lines from AA as well. These findings are corroborated with tumor micro-array data, and this is a strength. Overall the conclusion that Notch and GLI are increased in AA TNBC is well supported by the data presented.Weaknesses include the superficial nature of the description of the interaction between Notch and GLI1: it is unclear whether these proteins bind directly, and/or what the impact of this complex is on either Notch or GLI1 function. Additionally, the drug efficacy studies compare triple agents versus a single agent; it is not surprising that 3 agents work better than one.

We thank the reviewer for the comments. We have provided more data to show the interaction between Notch1 and GLI1 in AA-TNBC. We show that endogenous Notch1 protein can pulldown GLI1 in a coimmunoprecipitation assay and vice versa in AA-TNBC cells while no interaction is observed in WA-TNBC cells. Overexpression of GLI1 increases Notch1 transactivation. GLI1 overexpression increases the expression of Notch1-responsive genes and Notch1 overexpression increases the expression of GLI1-responsive gene. Together, these data show that GLI1 and Notch1 interact in AA-TNBC. We show that combined treatment with GLI inhibitor + Notch inhibitor results in more effective inhibition of AA-TNBC cell-derived tumors in comparison to Doxorubicin and Carboplatin combination. In contrast, WA-TNBC cells-derived tumors do not respond to combined treatment with GLI inhibitor + Notch inhibitor. These studies provide the rationale for examining the GLI1 + Notch inhibitor combination in AA-TNBC.